

# *In-situ* GPS records of surface mass balance and ocean-induced basal melt for Pine Island Glacier, Antarctica

David E. Shean[1,2], Knut Christianson[2], Kristine M. Larson[3], Stefan R.M. Ligtenberg[4], Ian R. Joughin[1], Ben E. Smith[1], C. Max Stevens[2]

[1]Applied Physics Laboratory Polar Science Center, University of Washington, Seattle, WA, USA
[2]Department of Earth and Space Sciences, University of Washington, Seattle, WA, USA
[3]Department of Aerospace Engineering Sciences, University of Colorado, Boulder, CO, USA
[4]Institute for Marine and Atmospheric research Utrecht, Utrecht University, Netherlands

*Correspondence to:* David E. Shean (dshean@uw.edu)

## Abstract

In the last two decades, Pine Island Glacier (PIG) experienced marked speedup, thinning, and grounding-line retreat, likely due to ice-shelf basal melt and marine ice-sheet instability. To better understand these processes, we analyzed 2008–2010 and 2012–2014 *in-situ* GPS records for PIG to constrain surface mass balance, firn compaction, and basal melt. We computed time series of horizontal velocity, strain rate, antenna height, surface elevation, and Lagrangian elevation change (Dh/Dt). The antenna height time series show a surface elevation increase of ~0.7–1.0 m/yr, which is consistent with model estimates for surface mass balance (SMB) of ~0.7–0.9 mwe/yr and ~0.7–0.8 m/yr downward velocity due to firn compaction. An abrupt ~0.2–0.3 m surface elevation decrease, likely due to surface melt, is observed during a period of warm atmospheric temperatures from December 2012 to January 2013. Observed Dh/Dt for all PIG shelf sites is highly linear, with trends of -1 to -4 m/yr and residuals of <0.4 m. Corresponding basal melt rate estimates range from ~10 to 40 m/yr, in good agreement with those derived from ice-bottom acoustic ranging, phase-sensitive ice-penetrating radar, and high-resolution stereo DEM records. The GPS and DEM records document higher melt rates within and near features associated with longitudinal extension (transverse surface depressions, rifts). Basal melt rates for the 2012–2014 period show limited temporal variability, despite significant changes in ocean heat content, suggesting that sub-shelf melt rates may be less sensitive to ocean heat content than previously reported, at least for these locations and time periods.

## 1   Introduction

The widespread availability of precise Global Positioning System (GPS) measurements has revolutionized the study of ice dynamics and glacier mass balance (e.g., Gao and Liu, 2001). Continuously operating dual-frequency GPS receivers provide high-frequency (1 Hz or less), highly accurate (< 1-3 cm) measurements of position, which can be used to derive surface velocity and elevation change. For applications involving ice dynamics, these measurements offer important constraints for the mass continuity equation, which equates surface elevation change with surface mass balance, basal mass balance, and ice flux divergence.

Surface mass balance (SMB) processes include precipitation, sublimation, wind redistribution of surface snow, and melt water runoff. Regional climate models forced by reanalysis output (e.g., RACMO, MAR) now provide daily to



monthly estimates of Antarctic SMB on a relatively coarse grid (~5.5 to 27 km). *In-situ* SMB measurements are, however, still essential for model calibration and validation. Traditionally, SMB can be measured using stake networks, automated weather stations, near-surface radar surveys, and firn/ice cores, all of which require substantial field operations in remote locations. These measurements also tend to bias model calibration towards locations that

are either accessible or of high scientific interest for other reasons. Reanalysis of glacier mass balance records indicate that these measurement biases can significantly affect results, often resulting in overestimates of cumulative balance, as dynamic areas are less well sampled (Andreassen et al., 2016). Antarctic firn/ice core records indicate that SMB variability over most of Antarctica during the last 800 years was statistically insignificant, but that accumulation increased more than 10% for high-accumulation coastal regions (e.g., the Amundsen Sea Embayment) since the 1960s

(Frezzotti et al., 2013). These areas are precisely those that that are most poorly sampled for SMB reanalysis calibration and validation using traditional methods (i.e., firn/ice cores).

Accurate knowledge of firn compaction and its spatiotemporal variability is essential for interpreting observed surface elevation change in remote sensing data (e.g., satellite altimetry), and for partitioning this change into components related to ice dynamics and SMB (e.g., Shepherd et al., 2012; Wouters et al., 2015). Depth-dependent compaction

rates can be estimated from a number of different methods, including vertical strain measurements (Arthern et al., 2010; Hamilton and Whillans, 1998), borehole optical stratigraphy (Hawley and Waddington, 2011), repeat phase-sensitive radio-echo sounding (pRES) measurements (e.g., Jenkins et al., 2006) and ice-penetrating radar observations of internal layers over time (e.g., Medley et al., 2014, 2015). In the absence of *in-situ* measurements, dynamic firn models forced by modelled SMB can provide estimates of compaction rates throughout the firn column, which can be

integrated to obtain estimates for surface elevation change over time (e.g., Ligtenberg et al., 2011).

Basal mass balance (BMB) beneath ice shelves (i.e., bottom melting, accretion) is driven by complex ice-ocean interaction. State-of-the-art ice-shelf cavity ocean circulation models offer some insight into sub-shelf ice-ocean interaction, but these models lack validation, as *in-situ* hydrographic observations are limited, especially within the sub-shelf cavity and the ice-ocean boundary layer. Some direct measurements are available from autonomous

submersibles (e.g., Dutrieux et al., 2014) and instrumentation packages deployed through ice-shelf boreholes (e.g., Stanton et al., 2013), but available data are limited to short time periods and small spatial extents. Precise measurements of surface elevation change from remote sensing observations (e.g., laser altimetry, digital elevation models (DEMs)) can also be used to infer BMB (e.g., Dutrieux et al., 2013; Pritchard et al., 2012; Shean, 2016), but time intervals between repeat observations are typically several months to years.

Here, we use temporally dense *in-situ* GPS records from the Pine Island Glacier ice shelf to constrain local SMB, firn compaction, flux divergence, and BMB. We use changes in observed GPS surface elevation and antenna height to validate SMB and firn model output, and then use firn model results to estimate the time-variable downward velocity of the GPS poles. Flux divergence is estimated from observed strain rates between GPS stations. These constraints are then used to isolate elevation change due to local BMB. This approach yields temporally dense estimates of basal melt

rates at spatially sparse GPS locations, which are combined with high-resolution DEMs from the same time period to provide spatial context for the observed elevation change. These complementary results for the PIG ice shelf provide new information about the time-variable magnitude and spatial distribution of basal melting, offering indirect



observations of ice-ocean interaction and BMB sensitivity to ocean heat content variability, with implications for other rapidly evolving "warm-cavity" Antarctic ice shelves.

## 1.1    PIG background

Pine Island Glacier is one of the largest and most dynamic ice streams in West Antarctica. Since the 1970s, PIG has experienced ~30 km grounding line retreat along its centerline (Rignot et al., 2014) (~8 km average retreat across full width of fast-flowing trunk (Joughin et al., 2016)), a ~75% increase in surface velocity (Mouginot et al., 2014), and >100 m of thinning (Bindschadler, 2002; Pritchard et al., 2009), with accelerated retreat beginning in the 1990s due to increased ocean heat content, circulation, and basal melt (Jacobs et al., 2011). Present-day surface velocities are ~4 km/yr, with annual discharge of ~130 Gt/yr (Medley et al., 2014; Mouginot et al., 2014) and mass loss estimates of 40 to 50 Gt/yr (Medley et al., 2014; Rignot, 2008). This mass loss is responsible for ~0.11 mm/yr global sea level rise (SLR), or approximately 40–45% of the total ~0.26 mm/yr Antarctic SLR contribution (Church et al., 2013; Rietbroek et al., 2016).

Figure 1 shows the fast-flowing portion of the PIG ice stream, which terminates in a large ice shelf ("main shelf") that is ~25 km wide, ~100 km long, and ~1–1.5 km thick across the grounding line. Basal melting accounts for ~70–80% of mass loss from the ice shelf, with estimated 2003-2008 melt rates of ~95–101 Gt/yr (Depoorter et al., 2013; Rignot et al., 2013) and 2008-2015 melt rates of ~80-90 Gt/yr (Shean, 2016).

The main shelf has complex surface topography, including km-scale surface ridges and troughs that correspond to basal keels and channels, respectively (Bindschadler et al., 2011; Vaughan et al., 2012). A series of longitudinal (along-flow) ridges and troughs are present along the shelf centerline, with transverse (across-flow) ridges and troughs along the lateral margins (Figure 1). Local basal melt rates vary considerably across these features (Dutrieux et al., 2013; Shean, 2016).

Hydrographic observations seaward of the PIG calving front in Pine Island Bay suggest that basal melting declined by ~50% between 2010 and 2012 (Dutrieux et al., 2014). Long-term 2009–2015 mooring records seaward of the southern calving front (Figure 1) show a significant decrease in ocean temperature (~1–1.5°C) over ~450–770 m depths from late 2011 to early 2012, and then again from mid-2012 to early 2013 (Christianson et al., 2016). These observations show that the ocean heat content at the PIG ice-shelf front varies considerably over monthly to interannual timescales.

## 1.2    PIG GPS sites

Several long-term GPS stations were installed on the PIG shelf as part of a larger investigation involving ice-sheet and ice-shelf dynamics (Bindschadler et al., 2011; Stanton et al., 2013). During the early part of this effort, two GPS stations continuously collected data from January 2008 to January 2010 on the southern PIG ice shelf (PIG2) and the fast-flowing, grounded ice upstream of the grounding line (PIG1) (Figure 1). In addition, a ~2x2 km array of five stations (SOW1–4, BOAR, Figure 2) was installed ~45 km downstream of the grounding line on the main shelf from January 2012 to late December 2013.



The stations used dual-frequency Trimble NetRS GPS receivers (2008–2010 sites) and NetR9 receivers (2012–2014 sites), with Trimble Zephyr Geodetic 2 antennas mounted on 12-foot (3.66 m) poles driven into the snow by hand, with pole bases set ~0.5–1.0 m beneath the surface [*Truffer*, personal communication, 2016].

High-resolution optical imagery and DEM data (section 3.6) over the 2012–2014 sites show that SOW1, BOAR, and
SOW3 were oriented approximately along a flowline within a longitudinal surface trough (Figure 2) that overlies a longitudinal basal channel. Transverse ice-penetrating radar profiles across this channel provide ice thickness estimates of ~450–460 m near the apex and ~540 m over adjacent keels (Stanton et al., 2013). Figure 2 shows estimated hydrostatic ice thickness for longitudinal and transverse profiles across the GPS array.

A borehole was drilled through the ice shelf, and an instrument package with an upward-facing ice-bottom altimeter
(acoustic ranger) was deployed beneath the shelf from January to February 2012. Measurements from this bottom altimeter and complementary pRES experiments provided basal melt rate estimates of ~14–25 m/yr within the longitudinal channel (Christianson et al., 2016; Stanton et al., 2013).

The 2012–2014 GPS array was located near several transverse surface depressions (Figure 2). Local surface slopes were ~0.6–0.9° within the largest of these depressions, immediately downstream of SOW3 and SOW4. A notable
linear depression located approximately 1 km upstream of SOW1 (arrow in Figure 2) opened as a rift in ~2014 (R1 in Jeong et al. (2016)), and was subsequently the site of a large iceberg calving event that occurred around July 2015. The fortuitous placement of the 2012–2014 GPS array near these features complicates interpretation of GPS records, but also provides new constraints on the spatiotemporal evolution of longitudinal strain and rift formation for the PIG shelf.

**2    Methods**

**2.1    GPS position/velocity processing**

As described in Christianson et al. (2016), GPS data were processed using differential-carrier-phase positioning relative to bedrock GPS sites (Backer Island [BACK, -74.26°N, -102.28°E] for 2012–2014 records; Howard Nunatak [HOWN, -77.31°N, -8.65°E] for 2008–2010 records) with epoch-by-epoch zenith tropospheric delay estimation.
Daily-averaged positions of these base stations were calculated using GAMIT and stabilized relative to a fixed circum-Antarctic reference frame using a Kalman filter (GLOBK, (Herring et al., 2015)). Geodetic GPS positions relative to WGS84 ellipsoid were calculated every 30 seconds; we analyzed a subset of these positions sampled at 10-minute intervals. All position data were converted to a local Cartesian horizontal coordinate system with orthometric heights above the EGM2008 geoid (Pavlis et al., 2012). Positions with uncertainty >8 cm were removed. The BOAR record
was curtailed on April 29, 2013 (1.31 years duration), when an abrupt ~2.0 m elevation decrease and corresponding horizontal offset occurred, suggesting that the pole fell over.

Horizontal velocities for each GPS station were computed from daily mean positions. Relative distances between stations were used to calculate strain, with linear fits to estimate strain rates.




## 2.2 GPS corrections

We estimated vertical tidal displacement for all GPS positions on the PIG ice shelf using the CATS2008A model (Padman et al., 2002). We used mean sea level pressure values from the 0.75°-grid-cell ERA-Interim reanalysis products (Dee et al., 2011) to correct for vertical displacement due to the inverse barometer effect (e.g., Padman et al.,
2003). To do this, we removed the 2002–2016 median (985.21 hPa) from 6-hour sea level pressure and scaled the residuals by ~1 cm/hPa.

Figure 3 shows that tidal amplitudes for the GPS sites range from approximately -0.9 to +1.3 m and IBE amplitudes range from -0.3 to +0.3 m. These signals were removed from the GPS antenna phase center elevations, and residual high-frequency noise was removed with a low-pass filter (1.5-day cutoff), yielding smoothed time series for further
analysis (Figure 3).

The constant 3.66 m pole length and 5.32 cm antenna phase center offset was removed from filtered absolute antenna elevation ($h_a$) to estimate corresponding absolute pole-base elevation ($h_{pb}$) relative to the EGM2008 geoid. Figure 4 shows a schematic of this geometry with additional observables as described below.

## 2.3 GPS antenna height and surface elevation

We computed mean daily antenna height above the surface ($z_a$) from L1 C/A code multipath surface reflections using the GPS interferometric reflectometry methodology outlined in Larson et al. (2015) (Figure 4). This method takes advantage of the fact that the interference between the direct and reflected GPS signals produces characteristic frequencies in signal-to-noise ratio data recorded by the GPS receiver; these frequencies are directly related to the distance between the GPS antenna phase center and the reflecting surface. Geodetic antennas are designed to suppress
multipath, so these interference patterns are best resolved at low GPS satellite elevation angles.

Antenna height solutions were calculated for elevation angles of 5–25°, which sample the surface within a radial extent of ~5–50 m. Local surface slopes at each site are negligible, eliminating the need for an azimuthal correction (e.g., Larson and Nievinski, 2013).

Antenna height ($z_a$) was subtracted from filtered absolute antenna elevation ($h_a$) to obtain daily records of absolute
surface elevation $h$ (i.e., firn-air interface) for all sites. The absolute antenna elevation accuracy was ~1 cm and daily antenna height precision was ~1 cm (Larson et al., 2015), with resulting surface elevation accuracy of ~1-2 cm. These surface elevation values are directly comparable with satellite/airborne laser altimetry data and stereo DEM products. The multipath antenna height above surface ($z_a$) was also used to estimate pole-base depth below the surface ($z_{pb}$) at each site.

Continuous antenna height time series were generated for all seven PIG GPS sites. The SOW3 record was curtailed on August 22, 2013, when antenna heights decreased below the ~0.5 m minimum threshold (Nievinski, 2013).

## 2.4 SMB and Temperature data

We analyzed estimates of 1979–2015 monthly and 2010–2013 daily SMB for three 27-km grid cells over the PIG shelf from the regional atmospheric climate model RACMO v2.3 (Ettema et al., 2009; Lenaerts et al., 2012; Van
Meijgaard et al., 2008; Van Wessem et al., 2014). The average 1979–2015 SMB ($\bar{a}$) is 0.91 mwe/yr for the grid cell



closest to the 2012–2014 GPS array (-75.07°N, -100.80°E, Figure 1). The values for adjacent grid cells are 0.74 mwe/yr near the grounding line of the main shelf (-75.15°N, -99.88°E) and 0.84 mwe/yr over the south shelf (-75.30°N, -101.14°E), providing some information on large-scale spatial variability. These values are consistent with SMB estimates of ~0.5–1.0 mwe/yr derived from CReSIS Snow Radar data collected upstream of the PIG grounding

line (Medley et al., 2014, 2015) and SMB estimates of 0.99 and 1.06 mwe/yr for stake measurements near 2006–2008 GPS sites over the upstream PIG trunk (Scott et al., 2009).

We also analysed 2011–2015 temperature data with 3-hour interval from the Evans Knoll (-74.85°N, -100.40°E, Figure 1) automated weather station (AWS) (Lazzara et al., 2012), located at an elevation of ~178 m (height above EGM2008 geoid) on a bedrock outcrop approximately 40 km north of the 2012–2014 GPS array (Figure 1). The AWS

temperature values were scaled to the surface elevation of the GPS array (~66 m above EGM2008 geoid) assuming a dry lapse rate of 9.8° C/km.

To provide historical context, we extracted 2-m air temperature from 1979–2015 for grid cells over the PIG shelf in 0.75°-resolution ERA-Interim reanalysis products (Dee et al., 2011) with 6-hour interval. The ERA-Interim temperature data display a bias of -2.81°C (median of offsets) compared to the scaled 2011-2015 AWS data, which is

consistent with previous evaluations (Jones et al., 2016). This bias was removed from the ERA-Interim temperature record, with residual offsets displaying root-mean-square error (RMSE) of 3.26 °C and normalized median absolute deviation (NMAD) of 2.77 °C.

### 2.5 Firn densification model

Model SMB output from RACMO2.3 was used to force the semi-empirical 1-D IMAU-FDM dynamic firn model

(Ligtenberg et al., 2011) with 3-hour timesteps. The IMAU-FDM output is available at 2-day intervals. Velocities ($v_{ice}$) across the firn-ice transition (defined as the layer with 917 kg/m$^3$ density) were assumed to be in equilibrium with average 1979–2015 SMB ($\bar{a}$). Vertical velocity components for surface accumulation, surface sublimation, surface snow drift erosion/deposition, surface melt, dry firn compaction ($v_{fc}$), and a vertical mass difference buoyancy correction were computed for the 2008–2010 and 2012–2014 periods. These estimates were combined to provide time

series of simulated surface ($\tilde{h}$) and pole-base elevations ($\widetilde{h_{pb}}$) for each GPS station.

### 2.6 Basal melt rate

Mass conservation for a column with ice-equivalent thickness H (after removing a thickness correction $d$ that accounts for total air content in the firn column) relates Eulerian thickness change ($dH/dt$) with dynamic thinning/thickening due to local flux divergence ($\nabla \cdot H\mathbf{u}$, positive for extension), surface mass balance $\dot{a}$ (meters ice equivalent), and

basal mass balance $\dot{b}$ (meters ice equivalent, defined as positive for melt):

$$\frac{\partial H}{\partial t} = -\nabla \cdot (H\mathbf{u}) + \dot{a} - \dot{b} \tag{1}$$

This approach assumes that the total firn air content ($d \approx 12$ m for the PIG shelf (Shean, 2016)) remains constant for the period $dt$, which precludes the need to account for thickness change due to firn compaction, while allowing for




variable ice-equivalent SMB input $\dot{a}$. This assumption is supported by the limited variability (~1-3% or +/-0.3 m) in modelled IMAU-FDM total firn air content for the three PIG shelf grid cells during the relevant ~2-year study periods. The material derivative definition relates Eulerian (fixed reference frame) and Lagrangian (reference frame moving with the ice column) thickness change:

$$\frac{DH}{Dt} = \frac{\partial H}{\partial t} + \mathbf{u} \cdot (\nabla H) \qquad (2)$$

5 Rearranging Equation 2 and substituting into Equation 1, we obtain the mass conservation equation for Lagrangian thickness change:

$$\frac{DH}{Dt} = -H(\nabla \cdot \mathbf{u}) + \dot{a} - \dot{b} \qquad (3)$$

For a floating ice shelf in hydrostatic equilibrium, we can estimate ice-equivalent freeboard surface elevation $h_f = (h - d)$, where $h$ is measured surface elevation and $d$ is firn air content. We can then estimate total ice-equivalent thickness H:

$$H \approx h_f \left( \frac{\rho_w}{\rho_w - \rho_i} \right) \qquad (4)$$

10 Assuming a constant density for ocean water ($\rho_w$=1026 kg/m³) and ice ($\rho_i$= 917 kg/m³), we can substitute Equation 4 into Equation 3, and rearrange to estimate basal melt rate from observed Lagrangian elevation change:

$$\dot{b} = -\left( \frac{Dh_f}{Dt} + h_f (\nabla \cdot \mathbf{u}) \right) \left( \frac{\rho_w}{\rho_w - \rho_i} \right) + \dot{a} \qquad (5)$$

Equation 5 depends on ice-equivalent surface elevation change. We now consider elevation change for the GPS pole base within the firn column of this simplified ice shelf.

The short-term effects of surface processes (e.g., accumulation, melting) are largely absent below the upper few meters 15 of the firn column. Thus, the elevation change rate of the GPS pole base ($Dh_{pb}/Dt$) is much more sensitive to compaction within the underlying firn. The downward velocity of the pole base due to firn compaction ($v_{fc}$) varies as a function of pole-base depth $z_{pb}$ within the firn column. Values for $v_{fc}$ can be estimated by integrating firn model compaction rates from the firn-ice transition to $\widetilde{h_{pb}}$ at each timestep. If SMB for the time period $dt$ ($\dot{a}$) is approximately equal to the long-term average SMB ($\bar{a}$), then:

$$\frac{Dh_{pb}}{Dt} \approx \frac{Dh_f}{Dt} + v_{fc} \qquad (6)$$

20 which can be substituted into Equation 5 to estimate basal melt rate from observed pole-base elevation change:

$$\dot{b} = -\left( \frac{Dh_{pb}}{Dt} - v_{fc} + h_f (\nabla \cdot \mathbf{u}) \right) \left( \frac{\rho_w}{\rho_w - \rho_i} \right) + \dot{a} \qquad (7)$$

We neglect the slight reduction in total ice-equivalent thickness at the pole base ($h_{pb}$) vs. surface ($h$), as pole-base depth is negligible compared to total ice thickness ($z_{pb} \ll H$). For the local flux divergence term, we use local freeboard thickness (as in Equation 7) or H from radar measurements (Stanton et al., 2013). Uncertainty is estimated as ~0.15 m/yr for downward firn-compaction velocity (Ligtenberg et al., 2011), ~0.1 m/yr for elevation change 25 associated with local flux divergence, and ~5 kg/m³ for the density of ice and ocean water.



If modelled pole-base velocities are correct, then basal melt rate estimates from $Dh_f/Dt$ (Equation 5) and $Dh_{pb}/Dt$ (Equation 7) should be similar. We also note that inferred basal melt rates are ~9 times more sensitive to pole-base elevation change ($Dh_{pb}/Dt$) and local $v_{fc}$ than SMB ($\dot{a}$) for a floating ice shelf.

## 3    Results

### 3.1    Velocity

Figure 5A shows the velocities of the PIG1 and PIG2 stations. On the floating ice at PIG2, surface velocity increased from ~355 m/yr to ~380 m/yr between 2008 and 2010 as the GPS migrated downstream, with increased speedup beginning in late 2008. Velocities at PIG1 increased at a relatively steady rate from ~420 m/yr to ~460 m/yr as the station moved toward the fast-flowing PIG trunk (Figure 1).

Figure 5B shows the 2012–2014 GPS velocities at SOW1, SOW2, BOAR, SOW3, and SOW4, which varied from ~3830–4040 m/yr (Christianson et al., 2016). Velocities at SOW1, BOAR, and SOW3 were similar, while SOW4 (closer to shelf centerline) consistently moved ~20 m/yr faster than these three sites, and SOW2 consistently moved ~15 m/yr slower. Thus, there appears to be ~30–40 m/yr dextral (right-handed) shear across the ~2.4 km distance between the SOW4 and SOW2 sites. This subtle transverse velocity gradient is also apparent in velocity mosaics (e.g., Christianson et al., 2016).

The velocity of all five stations varied by ~2–4% from 2012-2014, as described in detail by Christianson et al. (2016). In general, the stations displayed similar relative velocity evolution, with several abrupt >10–20 cm/day velocity changes (Figure 5).

### 3.2    Strain rate

For this analysis, we assume that SOW1, BOAR, and SOW3 GPS stations were oriented approximately along a flowline, and that the observed displacements represent longitudinal strain. Figure 6A shows that the observed cumulative displacement between SOW1 and SOW3 (initial distance 2073.0 m) was ~6.7 m (~3.4 m/yr), with extensional strain rates of ~0.0017 yr$^{-1}$. Observed strain rates between SOW1 and BOAR (~2.1 m/yr over 1045.2 m, 0.0020 yr$^{-1}$) were greater than those between BOAR and SOW3 (~1.5 m/yr over 1029.1 m, 0.0014 yr$^{-1}$). Transverse strain rates were relatively low, with compression (-0.0004 yr$^{-1}$) between BOAR and SOW2, and extension (0.0007 yr$^{-1}$) between BOAR and SOW4. Subtle changes in strain rates between SOW1 and SOW3 were observed from 2012–2014 (Figure 6C). In general, increased (decreased) extensional strain rates occurred between SOW1 and SOW3 following an increase (decrease) in absolute GPS array velocity.

These longitudinal strain rates correspond to shelf thinning rates of ~0.5–0.9 m/yr assuming no lateral spreading, with limited expected surface Dh/Dt (<0.07–0.13 m/yr) for measured ice thickness of ~450–460 m (Stanton et al., 2013) or for thickness estimated from local freeboard (~430–500 m).





### 3.3 Downslope flow

In addition to elevation change associated with strain rates and corresponding local flux divergence, some component of observed Dh/Dt may be related to deformation due to local gradients in the driving stress. Over grounded ice, this component of Dh/Dt is typically dominated by advection over basal topography, and the vertical component of the

corresponding motion ($V_0$) can be estimated using observed horizontal GPS displacement and local surface gradients from an independent DEM (e.g., Larson et al., 2015).

While this approach is irrelevant for a freely-floating ice shelf in hydrostatic equilibrium, we attempt to estimate an upper bound for the vertical component of Dh/Dt due to diffusion of local topography. To do this, we again consider observed relative horizontal displacements within the 2012–2014 GPS array. Local surface slopes near SOW1, SOW2,

and BOAR are negligible (<0.2°), so we assume $V_0 = 0$ for these stations. If all of the observed ~3.4 m/yr relative displacement between SOW1 and SOW3 was attributed to flow down ~0.6° local surface slopes at SOW3, then the associated $V_0$ magnitude would only be ~0.03 m/yr, which is negligible compared to the observed 5.2 m/yr Dh/Dt. Considering that much of the observed relative displacement appears to be related to longitudinal extension (Section 3.2), we neglect any potential Dh/Dt contribution related to $V_0$.

### 3.4 GPS antenna and surface elevation change

Figure 7 shows antenna elevation ($h_a$) change relative to the initial absolute antenna elevation ($h_{a0}$) at each station. These results show negative, highly linear ($R^2$ 0.98–1.00) $Dh_a/Dt$ (Table 1), with rates of -1.6 to -2.1 m/yr at SOW1, SOW2, and BOAR, and higher rates of -5.2 m/yr and -3.8 m/yr at SOW3 and SOW4, respectively. The PIG1 rates over grounded ice are -7.6 m/yr with apparent concave-downward curvature. This is consistent with $V_0$ expected from

local surface topography and dynamic thinning over the PIG trunk associated with velocity increases in 2006–2008 GPS observations (Scott et al., 2009) and satellite records (Joughin et al., 2010; Mouginot et al., 2014).

Figure 9C shows corresponding surface elevation ($h$) change relative to initial surface elevation ($h_0$) at each station. The 2008–2010 surface elevation change at PIG2 is negligible (-0.13 m/yr). By contrast, surface elevations decreased significantly at all 2012–2014 sites, with rates of -0.9 to -1.3 m/yr for SOW1, SOW2 and BOAR, and rates of -4.1

m/yr and -3.0 m/yr at SOW3 and SOW4, respectively.

Residuals about these linear fits (Figure 9D+E) are small ($Dh_a/Dt$ RMSE of 0.095 m and Dh/Dt RMSE of 0.143 m for sites on the PIG shelf), with some seasonal to annual variability observed at all sites. We also note apparent abrupt (~days-weeks) elevation changes that occurred across all stations in the 2012–2014 array (e.g., June 2012).

### 3.5 Antenna height evolution

Assuming that the pole base remains fixed within its original firn layer (effectively behaving as an isochron tracer in the firn column), any observed decrease in antenna height above the surface ($z_a$ in Figure 4, Section 2.3) can be attributed to surface accumulation (e.g, snowfall, deposition of snow by wind). Conversely, an increase in antenna height can be attributed to surface ablation (e.g., melt, sublimation, removal of snow by wind) and compaction of snow/firn above the pole base. For convenience, we define the reflector height ($z_{rh}$) to track cumulative accumulation

and ablation relative to the initial surface ($h_0$).



Initial antenna heights ($z_a$) were ~2.5 to 3.1 m above the surface, indicating that initial pole-base depths ($z_{pb}$) were ~0.6 to 1.2 m below the surface (Figure 8). Antenna height above the surface at all sites decreased over time (Figure 8), with observed $Dz_{rh}/Dt$ rates of ~1.0 m/yr for 2008–2010 sites and ~0.7–0.8 m/yr for 2012–2014 sites (with SOW3 at ~1.0 m/yr) (Figure 9).

At both PIG1 and PIG2, there are periods of relatively rapid reflector height increase (e.g., from May–August 2008), followed by a steady decrease (e.g., August 2008–February 2009). This is consistent with periods of increased snow accumulation followed by several months of ongoing firn compaction with limited snowfall. The 2012–2014 records show similar periods of abrupt increase and steady decrease, with more limited duration.

All records show an abrupt reflector height decrease (~0.2–0.3 m) between December 2012 and January 2013, which
is consistent with surface melting and/or enhanced firn compaction rates in the upper few meters of the firn column (see Section 3.7).

### 3.6    High-resolution DEMs

In addition to the GPS elevation data, we generated WorldView/GeoEye stereo DEMs (Shean et al., 2016) with 32-m posting over the PIG shelf (Shean, 2016) to provide spatial context for the GPS time series. A total of 7 WorldView
DEMs intersected the GPS positions between 2012–2014. We sampled DEM surface elevation at corresponding GPS positions and compared with GPS-derived surface elevation (as described in Section 2.3) where possible. Table 2 shows statistics for the sampled DEM elevation compared with GPS surface elevation at each site (Figure 8).

In general, we observe good agreement between the two datasets, with RMSE of 0.72 m and NMAD of 0.57 m for the full sample (n=25). The DEMs display a slight bias (+0.43 m) relative to the GPS surface elevation. We note that the
January 14, 2012 WorldView DEM is anomalously high, which biases DEM Dh/Dt trends with limited sample count (Table 2).

High-resolution Lagrangian Dh/Dt elevation-change maps (see methodology in Shean, 2016) were computed for the 2012–2014 GPS sites by forward propagating 32-m DEM pixels from two initial DEM products (February 2, 2012 and October 23, 2012) using interpolated, time-variable surface velocity maps from Joughin et al. (2010) and
Christianson et al. (2016). Lagrangian Dh/Dt maps were generated for all valid DEM combinations. Products with time interval between 0.5–2.5 years were aggregated, and median Dh/Dt values were assigned to initial DEM pixel locations.

We observe good agreement between DEM Dh/Dt trends (Table 2) and GPS surface Dh/Dt trends (Table 1), with values of -1 to -2 m/yr near SOW1, SOW2, and BOAR, and -4 to -5 m/yr near SOW3 and SOW4. The shorter DEM
Dh/Dt intervals (e.g., ~1 year for SOW1 and BOAR) display larger errors than longer DEM intervals (~2 years for SOW2 and SOW3).

Figure 10 shows the composite Lagrangian DEM Dh/Dt maps, which provide spatial context for the GPS Dh/Dt estimates. Little or no elevation change was observed over longitudinal ridges, while areas within and near transverse depressions experienced enhanced thinning (Figure 10). This thinning was concentrated on the upstream side of the
transverse depressions. The Dh/Dt products relative to the October 23, 2012 DEM (Figure 10D) also show the pattern of thinning associated with the rift that opened upstream of SOW1 in ~2014 (Jeong et al., 2016).





### 3.7 Surface mass balance

The 1979–2015 RACMO average SMB over the central PIG shelf is ~0.9 mwe/yr. Monthly SMB climatology shows low accumulation rates of ~0.01–0.04 mwe/month over the PIG shelf during the austral summer (November to February), and high accumulation rates of ~0.08–0.1 mwe/month during austral winter (March to October) (Figure

9F). Daily SMB products show periods of days to weeks with increased accumulation (e.g., March 2013) that can be correlated with abrupt increases in surface reflector height. By contrast, the reflector height records show a steady decrease due to ongoing near-surface firn compaction during extended periods with little/no accumulation.

The ~3-4 week period between December 24, 2012 and January 17, 2013 was relatively warm, with scaled AWS atmospheric temperatures of ~1–5°C for most days (Figure 9F). Surface elevations decreased by ~0.2–0.3 m across

the entire GPS array during this period (Figure 9A), which is consistent with significant surface melting. The daily RACMO SMB data also show two accumulation events during the last week of December 2012 (Figure 9E), which likely involve rain on snow. Some component of the observed surface elevation decrease could be associated with enhanced near-surface melting and firn compaction during these events. No corresponding short-term changes were recorded by the pole-base GPS elevations during the ~3-4 week period, suggesting that the processes responsible for

the observed surface changes were limited to the upper ~1.5 m.

An analysis of 1979-2015 ERA-Interim 2-m air temperatures over the PIG shelf shows many previous warm periods with greater magnitude and duration than the 2012/2013 period. However, the 2012/2013 warm period stands out in the past decade, with the most recent comparable periods during 2005/2006 and 2006/2007.

### 3.8 Firn model results

Estimated downward pole-base velocities due to firn compaction ($v_{fc}$) were all ~0.70–0.75 m/yr (Figure 7, Table 1) from 2008–2010 and 2012–2014, despite the range of initial pole depths. A slight decrease in the compaction rate occurred over time, but the curves appear linear at these depths (Figure 7). These values are consistent with observed $Dz_{rh}/Dt$ and $Dh/Dt$ during periods with little or no surface accumulation

Figure 9C shows that the IMAU-FDM simulated surface elevation ($\tilde{h}$) ranges from -0.1 to +0.4 m from 2008–2010

and -0.2 to +0.2 m from 2012–2014. The observed $D\tilde{h}/Dt$ trend is +0.17 m/yr from 2008-2010, with no significant trend from 2012-2014. The magnitude and timing of the modelled surface elevation variability is consistent with the detrended observed surface elevation change (Figure 9D). The observed $Dh/Dt$ trends (-1 to -4 m/yr) (Figure 9C), however, cannot be explained by modelled elevation change due to only SMB and firn.

### 3.9 Basal melt rates

We computed basal melt rates from surface Dh/Dt and pole-base Dhpb/Dt using Equations 5 and 7, respectively. The resulting values range from ~2-4 m/yr at PIG2 to ~40-43 m/yr at SOW3, with good agreement between the rates obtained by surface and pole-base elevation change (Table 1).

The 2012–2014 melt rate estimates show significant spatial variability. The three upstream stations (SOW1, SOW2 and BOAR) show lower melt rates of ~9–14 m/yr, while the downstream stations near the transverse depression

(SOW3 and SOW4) have higher rates of ~29–43 m/yr for the same time period.





## 4    Assumptions

The results in the Section 3 relied on several simplifying assumptions. We now offer further discussion of these assumptions and their potential influence on our results.

### 4.1    SMB spatial variability

We used modelled SMB from a single RACMO2.3 grid cell to drive the IMAU-FDM dynamic firn model, and applied the result to all GPS stations. We expect these parameters to vary spatially (e.g., Medley et al., 2015) due to local environmental conditions (e.g. PIG2 is >400 m higher than SOW1-4 stations on the shelf) and local surface topography (e.g., km-scale ridges/troughs), which will affect near-surface winds and snow redistribution.

In addition, the IMAU-FDM values do not account for horizontal advection of the 1-D firn column through spatially-
variable RACMO fields (accumulation, surface temperature, etc.) over time. The GPS sites over the PIG shelf are moving northwestward at ~4 km/yr (Figures 1 and 5), which is nearly double the observed PIG shelf velocities from the mid-1970s (Mouginot et al., 2014). Thus, the local firn columns beneath the GPS sites likely experienced variable SMB input over their ~50-100 km horizontal path during the corresponding 1979-2015 time period. This suggests that the true firn column thickness and compaction rates may differ from the IMAU-FDM estimates. For this reason, we
use a firn air content estimate ($d \approx 12$ m) derived from ice-penetrating radar two-way travel time and altimetry surface elevation measurements (see appendix A of Shean, 2016).

We also expect local SMB and firn column thickness variability within the array. The greater $z_{rh}$ surface reflector heights (a proxy for surface accumulation) at SOW3 indicate that enhanced local accumulation occurred within the transverse depression (Figure 2), likely due to wind-blown snow. However, we note that all of the remaining 2012–
2014 stations display similar $z_{rh}$ reflector height evolution, as do the two 2008–2010 stations (Figure 9A). Future RACMO and IMAU-FDM products with improved resolution should provide additional constraints on spatial variability.

### 4.2    Pole-base settling

Some of the reflector height increase and/or observed negative Dh/Dt could be related to settling of the poles within
the firn. We assume that the poles froze in place shortly after installation, and the contact area (~1200 cm$^2$ for a ~1-meter-long cylinder with ~3.8 cm diameter) with surrounding snow/firn should be sufficient to counter the downward gravitational force. Thus, we expect that the Dh/Dt recorded by the GPS antenna pole represents rates at the base of the pole, rather than rates within an overlying layer.

A related consideration involves heating of the exposed pole during summer, which might lead to decoupling from
the surrounding snow/firn and thus, additional penetration of the pole base within the firn. However, we do not see any indication of such settling from December 2012 to January 2013, when surface elevations decreased by ~20–30 cm and pole base elevations showed little change. The lack of pole-base elevation change suggests that surface meltwater did not percolate below ~1–2 m depth into the snow/firn.



### 4.3    Strain rate length scales

As discussed in Section 3.2, the expected surface $Dh/Dt$ from local flux divergence is <0.1 m/yr, assuming that the observed ~1.5 m/yr relative displacement is evenly distributed across the ~1 km distance between GPS stations (BOAR and SOW3). Even if this strain is concentrated over a shorter distance (e.g., ~200 m), this contribution only
increases to <0.5 m/yr, which is still significantly less than the observed ~3–4 m/yr $Dh/Dt$ signals. Interestingly, despite similar extensional strain rate estimates for SOW1-BOAR (0.0020 yr$^{-1}$) and BOAR-SOW3 (0.0014 yr$^{-1}$), we note a large difference in local $Dh/Dt$ values at these stations (Figures 9 and 10). This supports the assumption that local flux divergence for these sites is a minor component of the observed $Dh/Dt$, and consequently, that the observed $Dh/Dt$ is primarily caused by basal melt.

## 5    Discussion

### 5.1    Long-term SMB and firn compaction

The observed reflector height increase ($z_{rh}$) of ~0.7–1.0 m/yr at all GPS sites (Figure 9A) is consistent with observed long-term SMB estimates ($\bar{a}$) and downward near-surface velocity estimates due to firn compaction ($v_{fc}$). If we assume a near-surface mean snow/firn density of 0.5 kg/m$^3$, the ~0.7–0.9 mwe/yr RACMO SMB estimates correspond to a
surface elevation increase ~1.4–1.8 m/yr. Removing the expected ~0.7–0.75 m/yr surface elevation decrease associated with firn compaction, we arrive at expected relative reflector height increases of ~0.7–1.0 m/yr, which matches observed $Dz_{rh}/Dt$ values (Figure 9A+B).

The fact that we observe similar trends for surface $Dh/Dt$ and pole-base $Dh_{pb}/Dt$ (Table 1) also supports the assumption that 2008–2010 and 2012–2014 SMB ($\dot{a}$) is consistent with long-term 1979–2015 SMB ($\bar{a}$) and firn-compaction rates.
The limited variability in surface elevation at PIG2 (Figure 9C) suggests that the observed 2008–2010 SMB over the South PIG shelf was approximately in balance with ongoing firn compaction and basal melt. The 2012–2014 sites, however, show significant surface elevation decrease that greatly exceeds simulated IMAU-FDM surface elevation ($\tilde{h}$) variability (+/-0.2 m) due to SMB and firn compaction. No significant $D\tilde{h}/Dt$ trend is expected during this period, suggesting that SMB and firn compaction are consistent with long-term values, and the large observed Dh/Dt values
must be attributed to basal melt.

### 5.2    Residual Dh/Dt variability

The surface (Figure 9D) and pole-base (Figure 9E) residuals appear to be uncorrelated. This suggests that seasonal surface processes (e.g., accumulation influencing near-surface compaction rates) are not responsible for driving the observed pole-base variability. We considered several possible sources for the observed sub-annual variability in the
surface and pole-base records, including ocean (e.g., currents, sea surface height), atmospheric (e.g, pressure, temperature), and dynamic processes (e.g, resistive stress from sea ice and/or mélange in shear margins). Unfortunately, we were unable to definitively determine the cause(s) for these variations in the ~2 year GPS records.





Some of the short-term (days-weeks) variability (e.g. June 2012) observed across all five 2012–2014 stations (Figure 9D+E) could be related to insufficient or incorrect IBE correction. The magnitude and timing of these systematic anomalies, however, suggests that they are likely related to grounding/ungrounding events (Joughin et al., 2016).

### 5.3   Strain rate history, rifting, and grounding evolution

The lateral shear across the GPS array is related to increased longitudinal extension closer to the PIG centerline, likely due to locally enhanced ductile deformation (i.e., "necking" (Bassis and Ma, 2015)) across transverse depressions, and/or expansion of basal/surface crevasses and rifts. The SOW3 station, which lies within a large transverse depression (Figure 2), displays a slight acceleration in negative elevation change (Figure 9E), potentially due to increased local extension within the depression.

The fact that the rift immediately upstream of SOW1 ultimately became the site of the 2015 calving event (Jeong et al., 2016) suggests that the velocities observed at the GPS array are not necessarily representative of the velocities near the PIG grounding line. An upstream regrounding event would slow ice upstream of the GPS array, initially resulting in increased extensional strain rates across the transverse rifts/depressions, followed by a velocity decrease at the GPS array. Conversely, an upstream ungrounding event would initially lead to decreased extensional strain rates

across the transverse rifts/depressions, followed by an increase in observed GPS velocities.

We suggest that an upstream regrounding event in ~June 2012 (Joughin et al., 2016) could be responsible for increased strain rates across the GPS array (Figure 6). Similarly, an ungrounding event in ~April 2013 followed by a grounding event in ~November 2013 could explain the decrease and subsequent increase in strain rates.

There is an abrupt ~10–20 cm pole-base elevation decrease at both SOW3 and SOW4 in late 2013, near the end of the

records (Figure 9E). No surface elevation information is available at SOW3 due to missing antenna height data, but a corresponding surface elevation decrease is observed at SOW4 (Figure 9D). These elevation decreases do not appear to be related to site servicing. Rather, these observations are consistent with relatively abrupt local extension within the transverse depression at SOW3 and SOW4, but not the upstream sites. The timing of this event corresponds with observed lengthening of the large rift (R1) upstream of SOW1 (Jeong et al., 2016), supporting the hypothesis that

relatively rapid, localized extension occurred across the transverse depressions and rifts during this period.

### 5.4   Comparison with *in-situ* basal melt rate observations

The GPS basal melt rate estimates (~9-14 m/yr for SOW1, SOW2 and BOAR sites) appear consistent with those from bottom altimeter (~14.7 m/yr from January–February 2012) and pRES (~15–25 m/yr) measurements of Stanton et al. (2013). These measurements provide some validation for the GPS results, but a direct comparison may be imprudent.

The Stanton et al. (2013) measurements were taken at a site approximately 1.34 km upstream of SOW1 (K. Riverman, personal communication, 2016), within the same longitudinal channel, but much closer to the site of the R1 rift (Figure 2A). Given the complex and evolving local surface topography, we also expect local variation in basal topography and associated melt rates during the 2012-2014 period. Furthermore, the altimeter sampled a ~5 cm diameter spot with unknown upstream/downstream orientation, approximately 30–40 cm from the edge of the 20 cm borehole. Aside

from local melt variability expected due to turbulent flow near the altimeter pole or borehole edge, the altimeter





provided a relatively small spatial sample compared to the GPS results, which are sensitive to changes in a column of ice with much larger spatial extent (10s to 100s of meters).

## 5.5 Basal melt rate spatial variability

The GPS records at SOW1, SOW2 and BOAR show similar Dh/Dt rates and residuals, which is consistent with their apparent orientation on the same "block" between transverse rifts/depressions. They are also located over the same set of longitudinal basal channels, and should be exposed to similar sub-shelf circulation.

The DEM Dh/Dt maps show enhanced surface elevation change rates, and thus higher basal melt rates, on the north side (left side in Figure 10) of longitudinal basal channels and on the upstream side of transverse depressions. The latter is consistent with increased Dh/Dt observed at the SOW3 and SOW4 sites, located on the upstream side of a transverse depression.

This relationship is potentially related to enhanced buoyant flow over increased basal slopes (e.g., Jenkins, 2011) and/or turbulence as water within the upstream meltwater channel first enters the "cavern" of the transverse depression. We also suggest that the transverse depressions may serve as conduits connecting flow between the longitudinal channels, potentially leading to increased circulation velocity and higher melt rates within the transverse depressions.

## 5.6 Implications for melt rate sensitivity to ocean variability

The 2012–2014 GPS data reveal subtle (~2–4%) velocity changes that appear correlated with observed variations in ocean temperature records from moorings (Figure 1) in Pine Island Bay (Christianson et al., 2016). Our analysis supports the alternative Christianson et al. (2016) hypothesis, that these velocity variations are primarily related to upstream grounding evolution (Joughin et al., 2016), and extension across a series of transverse depressions (Figure 6).

Both the GPS pole-base $Dh_{pb}/Dt$ and surface Dh/Dt fits are highly linear, with no significant variation in inferred basal melt rates for this 2-year time period. If sub-shelf melt rates beneath the GPS array were directly related to ocean heat content beyond the shelf front in Pine Island Bay, and had decreased by ~50% as suggested by Dutrieux et al. (2014), a significant change in observed Dh/Dt would be expected during this period. The lack of any significant Dh/Dt deviation suggests that melt rates at these sites are not significantly affected by observed ocean temperature variability. This finding suggests that either: 1) these sites are not representative of melt rates for the greater shelf (e.g., those near the grounding line), 2) the oceanographic measurements near the PIG ice front (Figure 1) are not representative of water circulating beneath these ice-shelf sites, or 3) local melt rates are less sensitive to the observed oceanographic changes than previously assumed.

## 5.7 Future work

High-resolution velocity maps derived from sub-meter imagery could potentially constrain local velocity divergence and length scales for observed strain between GPS receivers. In addition, seismic data from stations deployed near the GPS array and regional sites could help constrain the timing and location of rift propagation and grounding/ungrounding events.



It may be possible to further constrain firn-compaction rates, and thus long-term SMB, using relative layer thicknesses observed in CReSIS snow radar measurements (e.g., Medley et al., 2015) or *in-situ* pRES observations (e.g, Jenkins et al., 2006). However, airborne radar data over the PIG shelf suffer from clutter due to km-scale topography and crevasses, while the available intermittent pRES records (Stanton et al., 2013) likely lack the sensitivity to detect small

changes in firn layer thickness during the ~3-week observation period. These limitations highlight the value of long-term GPS records to constrain surface evolution where observations are sparse and model results are poorly constrained. Expanding the scope of our study to include the full archive of GPS data for the Antarctic and Greenland ice sheets would offer a valuable new dataset for ice-sheet SMB and model validation.

## 6    Summary and conclusions

We analyzed 2008–2010 and 2012–2014 GPS records for PIG. These data provide validation for surface elevation and Dh/Dt derived from high-resolution WorldView stereo DEM records, with sampled DEM RMSE of ~0.72 m and NMAD of ~0.57 m.

The GPS antenna height records document a relative surface increase of ~0.7–1.0 m/yr, which is consistent with modelled SMB of ~0.7–0.9 mwe/yr and downward firn compaction velocities of ~0.7–0.75 m/yr. An abrupt ~0.2–0.3

m surface elevation decrease, likely due to surface melt, is observed during a period of warmer atmospheric temperatures from December 2012 to January 2013. Observed longitudinal strain rates for the 2012–2014 GPS array are ~0.001–0.002 yr$^{-1}$, with negligible associated surface elevation change.

Observed surface Dh/Dt is highly linear for the PIG shelf sites, with trends of -1 to -4 m/yr and residual RMSE of ~0.1 m for the 2012-2014 sites. Similar estimates with reduced residual variability are obtained after removing

simulated downward GPS pole-base velocity due to firn compaction from GPS antenna elevation records. Estimated basal melt rates are ~10 to ~40 m/yr near the center of the fast-flowing PIG shelf, and ~2–4 m/yr for the southern shelf. These melt rates are similar to those derived from complementary *in-situ* instrument records (Stanton et al., 2013) and high-resolution stereo DEMs (Shean, 2016).

Both GPS and DEM records show higher melt rates within and near transverse surface depressions and rifts associated

with longitudinal extension. Basal melt rates for the 2012–2014 period show limited temporal variability, despite significant changes in ocean heat content at the ice front and likely in the ice-shelf cavity. Residual elevation change variability is likely related to upstream grounding/ungrounding events and variable strain rates across transverse depressions/rifts.

### Acknowledgements

D.S. was supported by a NASA NESSF fellowship (NNX12AN36H). K.C. was supported by NASA grants NNX16AM01G and NNX12AB69G and NSF grant 0732869. K.L. was supported by NSF AGS-1449554. S.R.M.L. was supported by an NWO ALW Veni grant (865.15.023). An NSF OPP grant to CReSIS (ANT-0424589) provided support for I.J. and additional support for D.S.. H. Conway provided useful feedback on an earlier version of this manuscript and discussions with P. Dutrieux helped guide interpretation. M. van Wessem, P. Kuipers Munneke, and





M. van den Broeke provided RACMO SMB products. The AWS data are available from the University of Wisconsin-Madison Automatic Weather Station Program (NSF ANT-1245663). We acknowledge the substantial effort required to obtain the GPS data used in this study, involving multiple field campaigns led by R. Bindschadler, D. Holland, and M. Truffer, with significant contributions from many others. We acknowledge GPS data collection and archiving provided by the UNAVCO Facility with support from NSF and NASA under NSF Cooperative Agreement No. EAR-0735156. Resources supporting the DEM generation were provided by the NASA High-End Computing (HEC) Program through the NASA Advanced Supercomputing (NAS) Division at Ames Research Center.

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





**Tables**

| Site | Time period | Days | $h_{a0}$ (m) | $z_{pb0}$ (m) | $Dz_{rh}/Dt$ (m/yr) | $Dh_a/Dt$, $Dh_{pb}/Dt$ (m/yr) | $v_{fc}$ (m/yr) | $Dh/Dt$ (m/yr) | $Dh_{pb}/Dt$ - $v_{fc}$ (m/yr) | Surf. $\dot{b}$ (m/yr) | Pole-base $\dot{b}$ (m/yr) |
|---|---|---|---|---|---|---|---|---|---|---|---|
| PIG1 | 2008-1-13, 2009-9-4 | 601 | 484.28 | 0.71 | 0.83 | -7.60[*] | -0.75 | -6.76[*] | -6.85[*] | -- | -- |
| PIG2 | 2008-1-10, 2010-1-27 | 747 | 76.09 | 0.78 | 1.02 | -1.12 | -0.74 | -0.13 | -0.38 | 2.1+/-1.0 | 4.5 +/-1.7 |
| SOW1 | 2012-2-10, 2013-12-23 | 714 | 67.78 | 0.61 | 0.67 | -1.81 | -0.75 | -1.13 | -1.06 | 11.6+/-1.2 | 10.9+/-1.8 |
| SOW2 | 2012-2-10, 2013-12-23 | 714 | 64.72 | 0.89 | 0.75 | -2.08 | -0.74 | -1.33 | -1.34 | 13.4+/-1.3 | 13.5+/-1.9 |
| BOAR | 2012-2-10, 2013-4-29 | 476 | 66.44 | 0.82 | 0.68 | -1.58 | -0.77 | -0.91 | -0.81 | 9.5+/-1.1 | 8.6+/-1.8 |
| SOW4 | 2012-2-10, 2013-12-24 | 714 | 69.66 | 1.08 | 0.76 | -3.76 | -0.73 | -3.00 | -3.03 | 29.2+/-2.1 | 29.4+/-2.5 |
| SOW3 | 2012-2-9, 2013-12-24 | 716 | 63.16 | 1.16 | 1.00 | -5.23 | -0.72 | -4.10 | -4.50 | 39.5+/-2.7 | 43.2+/-3.2 |

**Table 1: GPS station data and Dh/Dt results. Fields include initial antenna height $h_{a0}$ (meters above EGM2008 geoid), initial pole-base depth $z_{pb0}$, reflector height change $Dz_{rh}/Dt$, antenna elevation change $Dh_a/Dt$ (equal to pole-base elevation change $Dh_{pb}/Dt$), modelled downward pole-base vertical velocity due to firn compaction $v_{fc}$, surface elevation change Dh/Dt, and corrected pole-base $Dh_{pb}/Dt$ after removing $v_{fc}$. Corresponding ice-equivalent basal melt rates $\dot{b}$ computed for both the surface Dh/Dt and corrected pole-base $Dh_{pb}/Dt$ (equations 5 and 7, respectively). [*]Note: PIG1 values over grounded ice do not include correction to remove expected Dh/Dt due to advection along local surface slope ($V_0$).**

| Site | DEM n | DEM dt (days) | DEM Dh/Dt (m/yr) | GPS-DEM RMSE (m) | GPS-DEM mean (m) | GPS-DEM std (m) |
|---|---|---|---|---|---|---|
| SOW1 | 5 | 302* | -2.30 | 0.69 | -0.26 | 0.64 |
| SOW2 | 5 | 619 | -2.03 | 0.76 | -0.46 | 0.60 |
| BOAR | 4 | 302* | -1.69 | 0.86 | -0.55 | 0.66 |
| SOW4 | 5 | 368* | -3.35 | 0.75 | -0.61 | 0.44 |
| SOW3 | 6 | 619 | -4.32 | 0.54 | -0.30 | 0.45 |

**Table 2: Statistics for WorldView DEM accuracy from comparisons with measured GPS surface elevation data. Asterisks identify records with shorter DEM Dh/Dt time intervals.**





**Figures**

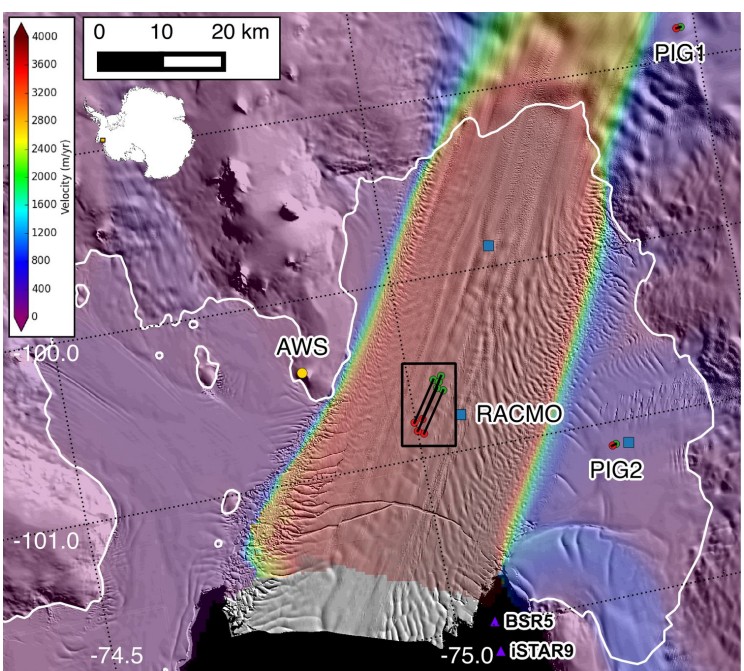

**Figure 1: Context for Pine Island Glacier ice shelf with 2006–2016 median surface velocity (Christianson et al., 2016; Joughin et al., 2010) over shaded relief map from October-December 2012 DEM mosaic. Black lines show ~2-year paths between initial (green) and final (red) GPS station locations. Yellow dot shows Evan's Knoll AWS and blue squares show RACMO grid cell centers used during analysis. Purple triangles beyond shelf front show locations of ocean mooring temperature data from Christianson et al. (2016). White line shows approximate 2011 grounding line (Shean, 2016). Black rectangle shows location of Figure 2A.**

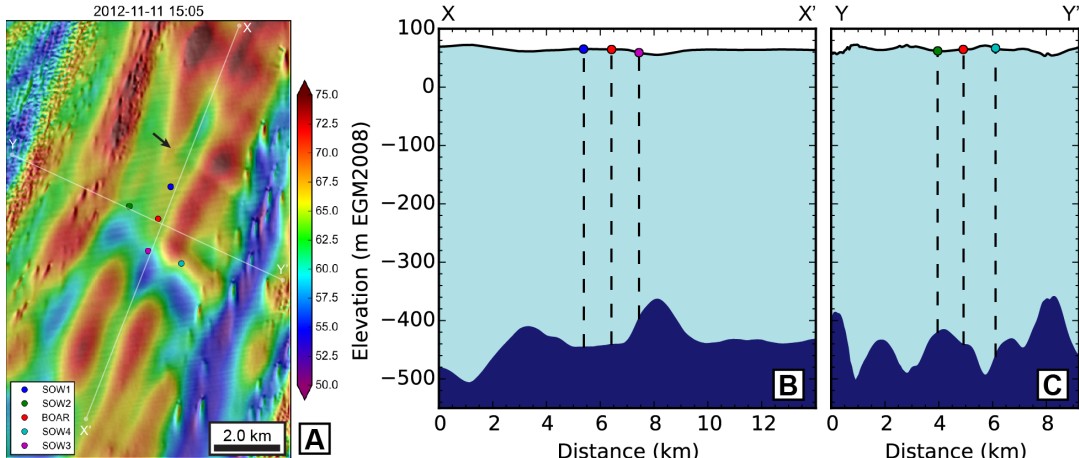

**Figure 2: A) WorldView DEM from November 11, 2012 with 2012–2014 GPS array positions overlaid. Note GPS positions relative to transverse depressions and location of R1 rift associated with 2015 calving event (arrow). Ice flow is from top right corner to bottom left corner. White lines show locations of profiles. B) Smoothed surface elevation (~0.5 km kernel, ~1·H) and freeboard thickness for longitudinal profile X-X' and C) transverse profile Y-Y'. Profile intersection is near BOAR (red point). Vertical exaggeration is 22x.**



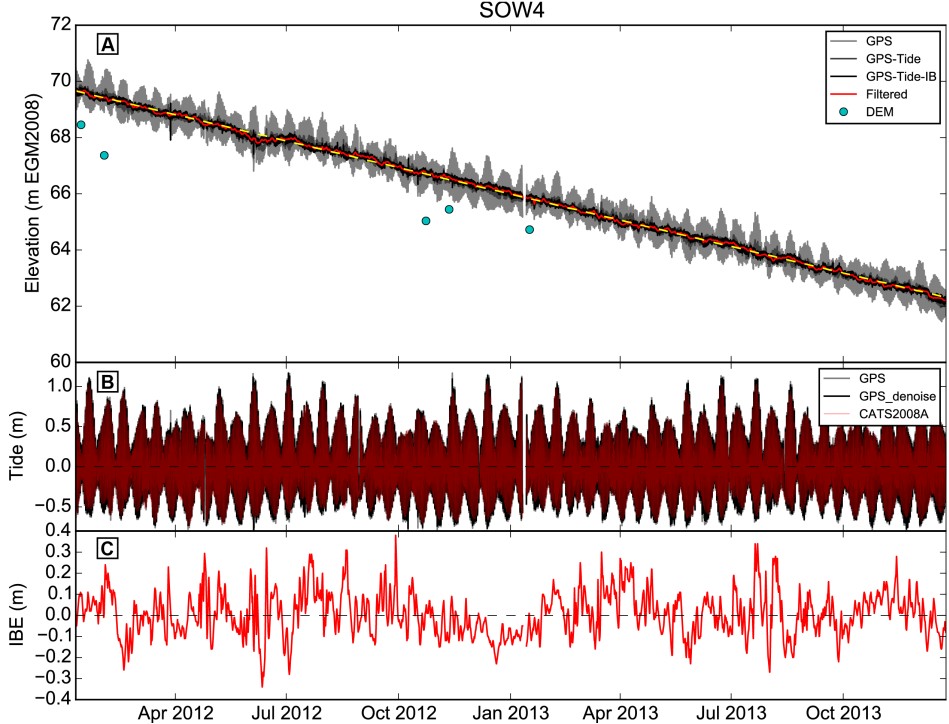

**Figure 3: A)** Original absolute GPS antenna elevation (light gray), after tide correction (mid-gray), and after tide+IBE correction (black) for SOW4. Red line shows smoothed time series and yellow dashed line is linear fit (-3.76 m/yr). Sampled DEM elevations (cyan) show surface elevation, which is offset from GPS antenna elevation by antenna height (see Figures 4 and 8). **B)** High-frequency (<1.5 days) component of GPS record and CATS2008A tide model prediction, showing excellent agreement. **C)** Estimated Inverse Barometer Effect (IBE) magnitude from scaled sea-level pressure.




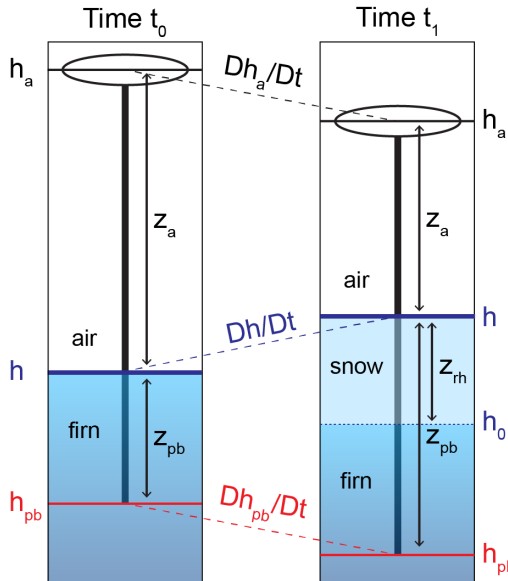

**Figure 4: Schematic of GPS station geometry. Absolute surface elevation (h, dark blue line) is computed by removing multi-path antenna height above surface ($z_a$) from absolute antenna elevation ($h_a$, black line). With known pole length, we can calculate pole-base depth below the surface ($z_{pb}$) and absolute pole-base elevation ($h_{pb}$, red line). At time $t_1$ (right panel), ongoing firn compaction resulted in decreased antenna and pole-base elevation, while new snow accumulation resulted in increased surface elevation. Reflector height ($z_{rh}$) is measured relative to the initial surface $h_0$ (dotted blue line).**



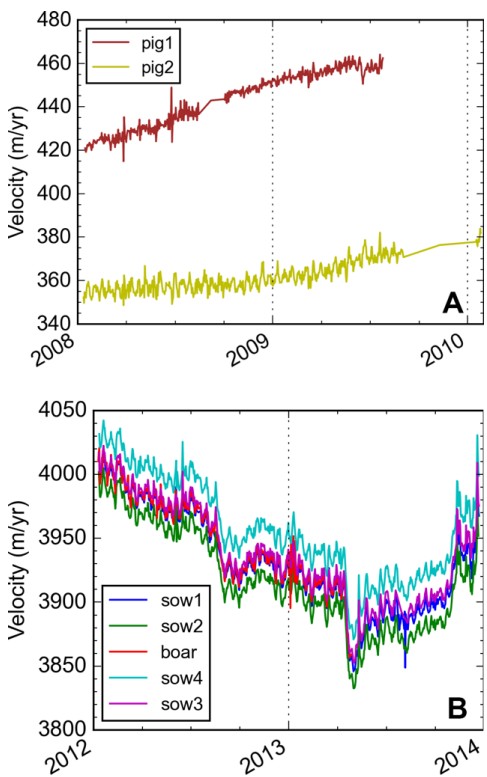

**Figure 5: Station velocities derived from daily mean positions for A) 2008–2010 GPS sites, and B) 2012–2014 GPS sites. Note offset between SOW2 and SOW4, indicative of lateral shear across the ~2 km wide array, with greater extension near the center of the PIG shelf.**



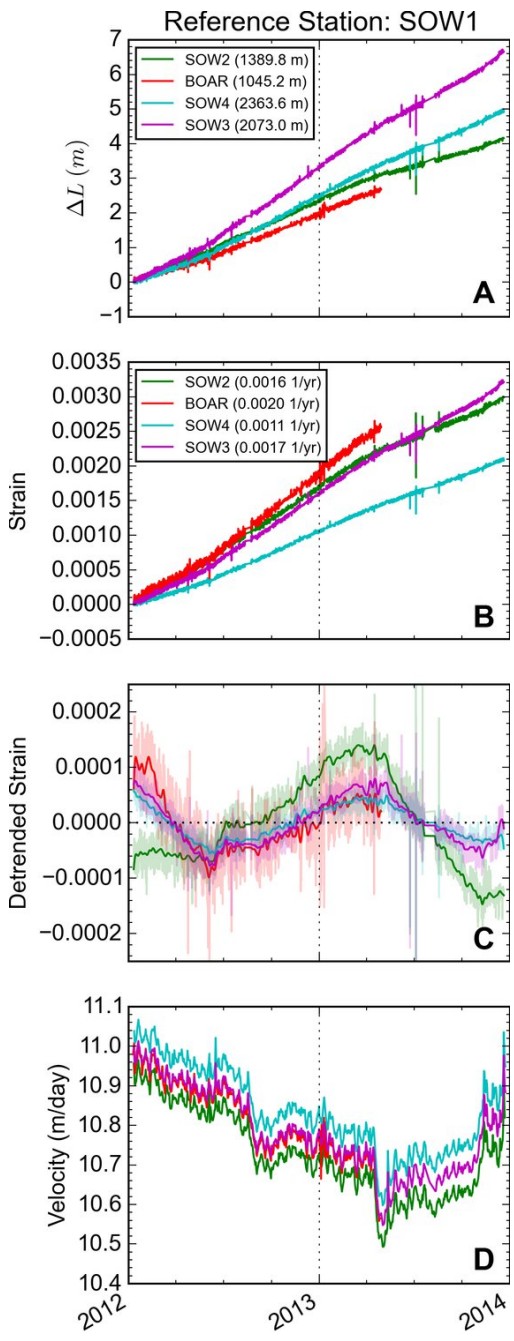

**Figure 6: A) Observed cumulative displacement between SOW1 and other 2012–2014 stations. Legend lists initial distances. B) Observed cumulative strain, with best fit strain rate listed in legend. C) Smoothed residuals from linear fit, highlighting subtle variations and inflections in strain rates. D) Daily GPS velocity. Note timing of abrupt absolute velocity changes observed at all sites and inflection points in observed strain across the array.**





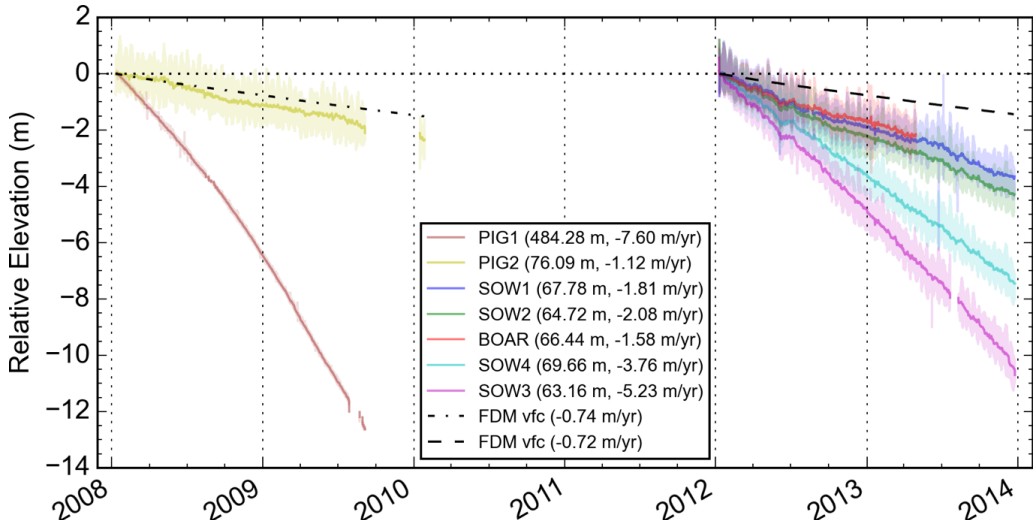

**Figure 7: Filtered (solid line) and original (transparent) GPS antenna elevation ($h_a$) relative to initial absolute antenna elevation (first item in legend). Legend includes linear $Dh_a/Dt$ fit to filtered antenna elevations. Dashed black lines show IMAU-FDM estimated downward velocity due to firn compaction ($v_{fc}$) at the pole base.**

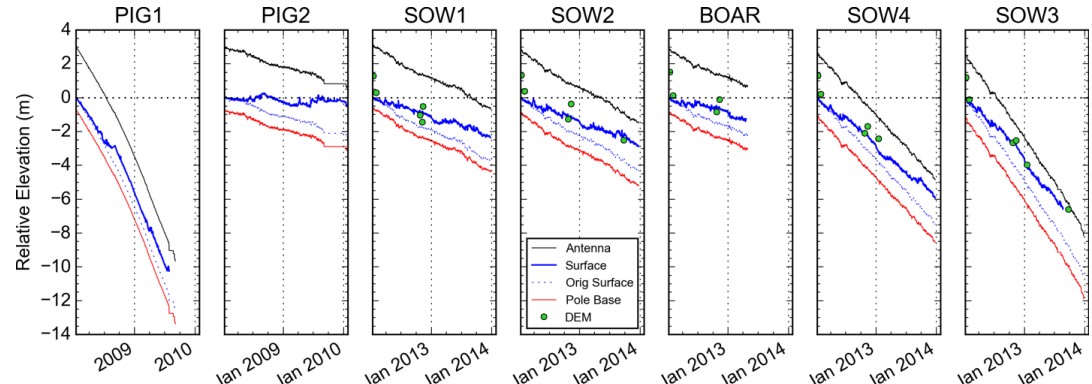

**Figure 8: Time series of GPS antenna elevation (black), pole-base anchor elevation (red), surface elevation (thick blue), and tracer for initial surface (dotted blue), all relative to the initial GPS surface elevation. See Figure 4 for schematic of different values. Green points show sampled WorldView DEM surface elevation, with observed DEM RMSE of 0.72 and NMAD of 0.57 m. Surface elevation decreased at all sites but PIG2.**





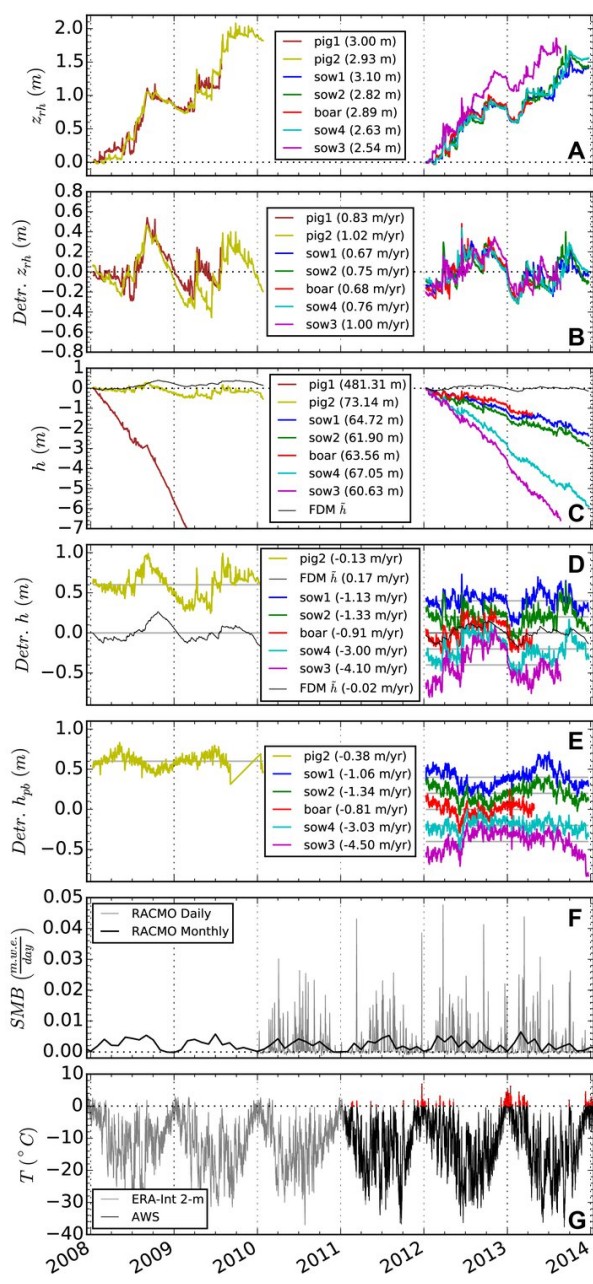

**Figure 9:** Plots of observed and modelled values for GPS sites (see Figure 4 for reference). A) Reflector height ($z_{rh}$) relative to initial surface ($h_0$). Positive values indicate surface height increase relative to GPS antenna. Legend indicates initial antenna height above surface. Note ~20–30 cm surface decrease from December 2012 to January 2013. B) Detrended reflector height ($z_{rh}$), with linear fit listed in legend. C) Surface elevation $h$, with initial surface height (m above EGM2008 geoid) in legend. Black lines show IMAU-FDM modelled surface elevation ($\tilde{h}$) due to SMB and firn compaction. Note that




due to basal melting, observed *dh/dt* for 2012-2014 sites is significantly greater than modelled $d\tilde{h}/dt$. D) Detrended surface elevation *h* and detrended IMAU-FDM $\tilde{h}$, with arbitrary y-axis offset. Legend lists linear fit (corresponding values listed in Table 1). E) Detrended GPS pole-base elevation ($h_{pb}$) after removing $v_{fc}$ (see Figure 7), with arbitrary y-axis offset. Legend lists linear fit, which can be directly compared with *Dh/Dt* fits in D (see Table 1). Note reduced residual magnitude and dampened ~seasonal signal compared to D. Unlike surface records, no significant change is observed from Dec. 2012 to Jan. 2013. F) Daily and monthly RACMO2.3 SMB. G) Observed AWS and ERA-Interim 2-m T records for PIG shelf, with above-zero AWS T plotted in red. Note period of extended warm T in Dec. 2012 to Jan. 2013, which corresponds to ~0.2–0.3 m surface elevation decrease.

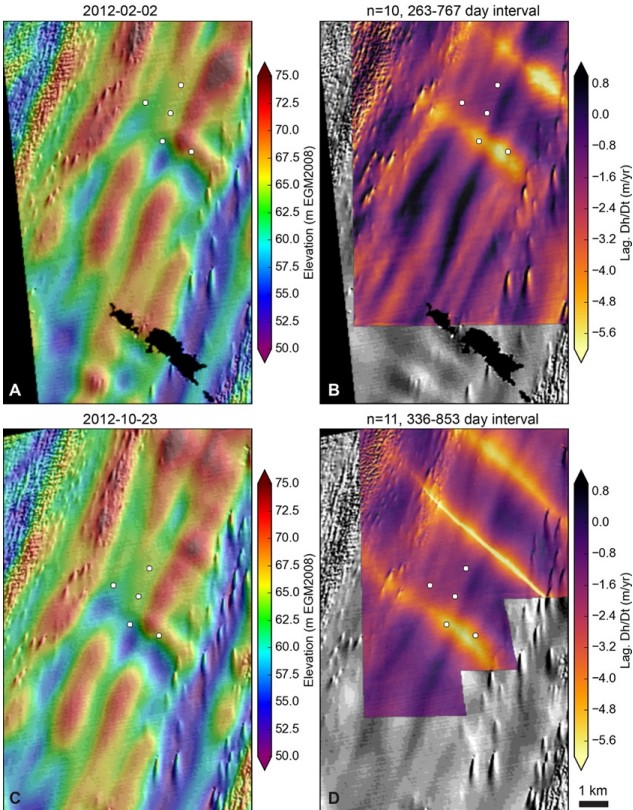

Figure 10: WorldView DEMs and composite median Lagrangian Dh/Dt products generated using A-B) initial DEM from February 2, 2012, and C-D) initial DEM from October 23, 2012. Note enhanced thinning observed within transverse depressions and rift upstream of GPS array. The Dh/Dt maps are used to calculate basal melt rates (scaling factor of ~9, e.g., ~1.0 m/yr Dh/Dt ≈ ~9–10 m/yr melt rate)