# Peer review of "In-situ GPS records of surface mass balance and ocean-induced basal melt for Pine Island Glacier, Antarctica"

_The Cryosphere, 2016_

## Referee Comment (RC1) · Anonymous Referee #1 · 6 Mar 2017

Review of:

**In-situ GPS records of surface mass balance and ocean-induced basal melt for Pine Island Glacier, Antarctica**

*By David E. Shean et al.*

This paper discusses time series measurements of surface elevation change made using an array of GPS receivers set out on the floating tongue of Pine Island Glacier. These and other data are used to quantify the processes contributing to changes in ice thickness, especially the surface and basal mass balance. The analyses are described in some detail and the results are interesting. The work is therefore certainly publishable, but my feeling is that more work should be devoted to the presentation to make the results more accessible. At present there is no clear logical flow from definition of the problem, through observation and analysis of observation and ancillary data, to conclusions. The discussion jumps between observations, processes, models, supporting data, and it is difficult to keep track of what has actually been observed and what the ultimate goal of the analyses is.

A bewildering range of surface height variables are introduced, many of which are not independent but simply derivatives of the others, and this really adds to the (unnecessary) complexity of the presentation. The problems are immediately apparent in that abstract. We are told that time series of "antenna height, surface elevation and Lagrangian elevation change (Dh/Dt)" are to be presented. But without any definitions the reader is left to guess what these variables actually are and how they are distinct. Reading the paper, it seems that "height" refers to measurements relative to the instantaneous snow surface, while "elevation" refers to measurements relative to the fixed geoid. However, "elevations" are confusingly denoted by the symbol "h", and the terminology is not used entirely consistently. The next sentence in the abstract says that "The antenna height time series show a surface elevation increase". If the increase is relative to the geoid, the ice must the thickening, right? Apparently not. Later on we are told that "observed Dh/Dt" is negative, implying thinning. This is probably the worst example of inconsistent terminology, but the discussion could be simplified throughout by reducing the number of variables to just the independent ones and a couple of critical derived ones. To add to the current confusion, the schematic defining all the variables (figure 4) actually shows Dh/Dt as positive.

The most fundamental variable measured by the GPS is the antenna elevation. A subsidiary measurement, and a really nice addition, is the height of the antenna above the snow surface. To avoid potential confusion between elevation and height (especially if the authors are not going to be entirely rigorous about their usage), I would recommend using an entirely different name for antenna height, perhaps "ground clearance" or "exposed pole length". Apart from these two variables, nothing is directly measured by the GPS. Everything else is a derived quantity, so why not just stick to these, unless it is absolutely essential to introduce something new. One derived variable that is worthwhile having is the snow surface elevation, since that is what the satellite stereo imagery (introduced later as an ancillary dataset) measures, but that is just the antenna elevation minus the exposed pole length. The "pole base elevation" is a useful reference level, but is just the antenna elevation, minus the fixed pole length, unless the pole angle relative to the vertical changed with time (a possibility that does not seem to have been considered). I don't see the point of introducing the "reflector height" variable which is just the initial "antenna height" (or ground clearance/exposed pole length) minus its initial value. So trends in "antenna height" and "reflector height" are identical, apart from the sign. Arguably the "pole base depth" below the surface (or

"buried pole length") is useful in that compaction rates above this level and below it ultimately need to be considered separately.

By the way, given that one pole fell over completely, can you be sure that your measurements are free of contamination from changes in the pole angle? Were there repeat measurements of pole angle? You also make the assumption that the pole base is the point that is fixed in the firn. That seems reasonable to me, but did you put an insulating stopper in the end of the pole? That would make your assumption more justifiable.

So for the GPS time series there are two measured quantities (antenna elevation and exposed pole length) and two derived quantities (surface elevation and buried pole length). I don't really see why you need any more. These can then be used to derive the surface and basal mass balance. The most consistent way of doing this would be to define a local vertical coordinate that is zero at the base of the pole and positive upwards. Then taking your equation (3) and substituting:

$$H = (z_{ia} - d_a^+) - (z_{io} + d_a^-)$$

where $ia$ is the ice-atmosphere interface, $io$ the ice-ocean interface and $d_a^{+/-}$ is the thickness of air above and below the origin, you get two separate equations for the surface and basal mass balance:

$$\frac{D}{Dt}(z_{ia} - d_a^+) = \dot{a} - (z_{ia} - d_a^+)\nabla.\boldsymbol{u}$$

$$-\frac{D}{Dt}(z_{io} + d_a^-) = -\dot{b} + (z_{io} + d_a^-)\nabla.\boldsymbol{u}$$

because the reference surface is assumed to be a material surface. Now, some of these terms you have implicitly ignored (probably perfectly valid, but it would have been helpful to see the process and justification of the simplifications), and of the others, some come from your GPS measurements and some from ancillary data and assumptions.

The divergence term comes from your measurements of the rate of change of the distances between GPS stations. Or does it? The methods section describes how you derive absolute positions and motion relative to fixed reference stations. But the absolute motion is unimportant here as your calculations are in a Lagrangian framework. Couldn't the inter-stake distances be derived more simply and accurately by differential positioning of one station relative to the other? No fixed base station is needed then. And given that you have all the inter-station distances as a function of time, why don't you make a direct estimate of the divergence from those data? Actually, since you have five stakes, and you only need three to do the calculation, you can do it with multiple stake combinations and get some idea about the spatial variability. Why haven't you done that? I found your ad hoc calculations based on the assumption that some stakes were oriented along flow, and that along-flow extension should dominate, to be rather unsatisfactory.

The air thickness terms come from the firn modelling, which needs a much clearer discussion of the inputs and outputs. I think it is driven by the "SMB and Temperature data" discussed in section 2.4. In which case, why is that section not a part of a larger section that discusses the firn model? As it stands, it is presented to the reader as a standalone estimate of SMB, but that is one of the quantities promised in the title from the GPS records. The point being, I assume, that you want regional estimates to drive the firn model, distinct from the point estimates that are the result promised in the title? How sensitive are your firn model results to these inputs? Basing your temperature on a measurement at Evans Knoll and an assumed lapse rate will introduce a warm bias. It is well known that a stable inversion layer can form over the flat topography of the ice

shelves, so surface temperatures can be lower than those on surrounding higher ground. I doubt that ERA-Interim captures this. Could you not use remote sensing data? The outputs from the firn model are never clearly presented. A vertical velocity associated with compaction is quoted in a number of places, but the separation of compaction into the components that occur above and below the pole base (as in the above equations is never made clear). These are key components of the results, and I would have expected to see some graphical presentation of them. The surface height from the firn model is shown, but this doesn't tell the reader what was actually used in the calculations of surface and basal mass balance.

The $z_{ia}$ term is simply the (assumed fixed) pole length minus the (measured) exposed pole length. The $z_{io}$ term is a bit more complex, since it must come from the measurements of antenna elevation and an assumption of isostatic equilibrium:

$$-\rho_w(z_{io} + h_0) = \rho_i(z_{ia} - d_a^+) - \rho_i(z_{io} + d_a^-)$$

where $h_0$ is the elevation of the reference surface above the geoid, that is the measured antenna elevation minus the fixed pole length. This leads to:

$$-(z_{io} + d_a^-) = \frac{\rho_w}{\rho_w - \rho_i}(h_0 - d_a^-) + \frac{\rho_i}{\rho_w - \rho_i}(z_{ia} - d_a^+)$$

which is now clearly in terms of the two directly measured quantities and two outputs from the firn model. Now the melt rate can be derived from the following equation:

$$\frac{\rho_w}{\rho_w - \rho_i}\frac{D}{Dt}(h_0 - d_a^-) = -\dot{b} - \frac{\rho_i}{\rho_w - \rho_i}\dot{a} - \frac{\rho_w}{\rho_w - \rho_i}(h_0 - d_a^-)\nabla.\boldsymbol{u}$$

In the above notation $h_0$ is the same as the authors' "pole base elevation", so I think a couple of the terms in this equation are the same as those in their equation (7), but I am not sure about the rest. I did not really follow all the assumptions made in the derivation of (7) and I would argue that the above is clearer in its relationship to the directly measured antenna elevation, the firn model output and the surface accumulation rate (itself derived above from a directly measured quantity and firn model output).

Of course, this equation and the authors' version relies on the assumption of isostatic equilibrium. Curiously the validity of this assumption is never discussed. However, the quoted thicknesses measured from radar differ from those calculated from surface elevation, so it is clear that the assumption does not hold. Herein lies the major weakness of the paper. It is well-known that over length-scales comparable with the ice thickness, vertical shear stresses can partially support the ice. This has been shown to be the case on similar channel-like features on a number of ice shelves (McGrath et al., 2012, Ann. Glaciol., 53, 10-18; Jenkins et al., 2006, J. Glaciol., 52, 325-346), including on Pine Island (Vaughan et al., 2012, J. Geophys. Res., 117, F03012). The key point is that while the ice is not freely floating (as it is not on the scale of the stake network discussed here) the regions of thinner ice will be sinking as the ice deforms under the vertical shear stress. So if you set up a GPS station in a channel you will see a surface lowering even if the ice thickness remains constant. Such a process could introduce a significant bias to the estimates of basal mass balance that does not appear to have been considered by the authors. It might, for example, explain why the surface elevation changes are rather steady despite the change in ocean conditions that have been inferred to have changed the melt rates.

Overall, my recommendation would be to restructure the paper along the following lines:

Introduction: A general introduction to the region and the problem to be addressed. Don't discuss your GPS sites in detail yet.

Method: Describe the principle of how measurements of the elevation and exposed pole length can be used to calculate the surface and basal mass balance. Introduce the reference frame and variables and show what additional information you need.

GPS data: Describe the set-up of the GPS sites, the data collected and the processing to get to the key results of antenna elevation and exposed pole length and inter-pole distances, and the uncertainties in all these numbers.

Divergence: Describe the calculations and show the horizontal variability. Discuss the uncertainties.

Firn compaction: Describe the model and the necessary input data. Show the outputs that you actually use in your calculations.

Results: Calculate surface and basal mass balance and uncertainties. Compare with other studies.

Discussion: Discuss all the other potential sources of error. Don't forget the impact of adjustment towards isostatic equilibrium. Not only is it a source of elevation change that is not related to thickness change, but it gives rise to a flow divergence that changes with depth, so your measurements of surface divergence are not representative of the depth-mean.

Conclusions: Can you say for sure that the melt rates were steady, given the uncertainties and potential biases? If it was, how do you explain that observation?

Finally, some more minor points:

I would recommend making it clearer in the title that the measurements are on the floating part of Pine Island Glacier and that the surface and basal mass balance estimates are derived from GPS measurements. The wording at present suggests that the results are directly measured by GPS.

Some of the figures could be improved. In particular, it would be much easier if the sites were labelled in Figure 2. The colour code is really hard to use. Again, label them on Figure 10.

Why did you use a tide model to remove the tides from your records? Why not do a harmonic analysis of the elevation data themselves? Your records are long enough to make reliable estimates of more tidal components than are given in the CATS output, aren't they?

In summary, while this is a long review, there are some really nice results contained in this paper. I also think the detailed discussion of the data and analyses could be a real strength, if it were presented in a more readily accessible way. The current presentation left me a little confused as to what the bottom line really is. The title promises estimates of both surface and basal mass balance, but the calculations are dependent on the results of a firn model that needs surface mass balance as an input. It is never made clear to what extent the author's regard their calculations of surface accumulation as independent results, or if the discussion is merely a demonstration that measured changes in exposed pole length are consistent with, rather than an addition to, prior knowledge. Likewise, with the basal mass balance. Do those results represent an addition to our knowledge of melting at the base of Pine Island, or are they more a demonstration of the limitations of using surface elevation data to infer melt rates at these spatial scales?

---

## Referee Comment (RC2) · L. Padman (Referee) · 2 May 2017

SUMMARY

This paper describes a GPS-based technique to make estimates of basal mass balance (BMB) and surface mass balance estimates for Pine Island Glacier ice shelf (PIGIS). The inclusion of GPS-derived snow surface height relative to the GPS antenna is a particularly interesting addition to standard ice-shelf GPS analyses, and the results are tied in well to other utilized datasets including the stereo-imagery of the DEM and the surface snow/firn state from a RACMO-based firn density model (FDM).

I have the advantage of being late in my review, so I don't need to make the detailed

review comments in RC1. However, I agree with all of these. In particular:

(a) It took me a long time to get the actual vertical coordinate system sorted out in my head. The "surface relative to the receiver" is a great addition, but it means finding clearer symbols, and explaining the coordinate system more carefully, early. (e.g., Already by line 15 in the Abstract, I was confused by the statement that "The antenna height time series shows a surface elevation *increase*" on an ice shelf that is thinning rapidly.) RC1 suggested fewer symbols. I agree with this, and suggest using relative height values as close to fundamental measurements as possible (the 'h' variables in Fig. 4), rather than derived difference quantities ('z' variables) even if it means adding text volume. Then also use longer but more descriptive subscripts. So, e.g., instead of $z_a$, use ($h\_antenna - h\_snow\_sfc$).

(b) The issue of whether the hydrostatic assumption is really appropriate in the presence of small-scale ice thickness variability needs to be better addressed. Maybe this has something to do with the obscure (to me) "diffusion of local topography"? But somehow convince us that hydrostatic is okay.

(c) I agree with RC1 on the general layout of the revised paper. Revisions responding to RC1 cautionary comments about data analyses (e.g., maybe other poles also lean?) run the risk of weakening your conclusions specifically about BMB. However, I don't think that matters too much, as your paper will still describe an interesting approach to GPS analysis that will likely lead to much improved GPS installation practices and data value from future GPS sites.

Specific comments are divided into "Major", where I'd like to know what you did in response, and "Minor", which don't require documentation in revision. Numbers refer to original page.line.

– Laurie Padman

MAJOR COMMENTS

Title is not right. First, what "*not* in-situ" method is available for GPS? Second, there are other important data sets and model output sets used here.

1.15: This is when it first became clear I didn't understand the vertical coordinate system; see SUMMARY comments and RC1.

3.7-8: The Jacobs et al. (2011) paper compares just one cruise in 1994 with another in 2009. This doesn't seem like robust evidence for a causal link between the "increased ocean heat content" and accelerated mass loss. It probably is, although glacier dynamics must figure into this, so maybe put an "attributed to" in there.

3.9-10: The 40-50 GT/a is *net* mass loss *from the grounded glacier*, right? Otherwise these numbers don't make sense given 130 Gt/yr discharge. Need to be specific.

6.13-25: What does a bias in the atmospheric model do to the results from a firn density model? And is the bias even taken into account when the FDM is run? Since RACMO2.3 is run from ERA-Interim (I think), then presumably RACMO2.3 has similar biases that need to be corrected for before running the FDM.

9.8: What is "diffusion of local topography"?

9.34: "reflector height" is a confusing term. See extended discussion of coordinate and naming issues in RC1.

10.35: Identify, on Fig. 10, the "flow" and "transverse" directions.

MINOR COMMENTS

3.9: either "with annual discharge of ∼130 Gt" or "with discharge of ∼130 Gt/year".

5.2-3: I agree with RC1 that it doesn't make sense to detide with a tide model when you have tide-resolving GPS. It's nice to see that CATS works well, so keep the comparison, but detiding with a tide model only makes sense for short or gappy GPS records. A better approach would be using T-Tide (in matlab) or the equivalent Foreman (1977) FORTRAN code.

5.25: Does "firn" include "snow"? Fig. 4 suggests that snow and firn are regarded distinctly. If so, the interface you are getting the relative surface elevation from is not necessarily "firn-air"

6.21: 917 kg/mˆ3 is solid ice, right? So it's a bad criterion for the firn-ice transition.

page 7 and elsewhere. Once the vertical coordinate symbol set is finalized, make sure all occurrences are formatted the same, e.g., \it{H}, not H.

7.25: 5 kg/mˆ3 seems like a high error for water density, and probably also high for solid ice, although there is might be used as a fudge factor for firn density profile.

8.14: "30-40 m/yr" doesn't seem "subtle" to me.

8.17: velocity units on Fig. 5 are m/y, so don't use cm/day in text.

9.22: (a) first call to Figure 9 is to Fig. 9C. Organize panels in the order that they come up. (b) Figure 9 called out before Figure 8?

10.5: either "from May to August" or "for the period May—August"

10.18 and 16.12: What is "NMAD" ?

10.25-27: Not sure what you mean by "aggregated" here.

10.34: delete "(Figure 10)"; sentence already starts "Figure 10 shows"

11.5: In text you quote mwe/month but in the figure you send us to (9F), SMB is given as mwe/day, so I can't directly compare them.

12.2: delete "the" in "The results in the Section 3"

14.33: repeat "bottom" in "bottom altimeter" here. I lost the thread of what altimetry you were talking about. Alternatively, reorganize this to tell us all about the altimeter then compare its results with your estimates.

15.11-14: This text could be tightened up.

FIGURES

See RC1 for some other figure comments.

F.1: Add a bold "North" (N) arrow since occasionally you refer to directions and it is hard to get oriented with this map.

F.2A: again, north arrow would be good, and a flow-direction arrow would help. (Better than just having it in the caption.)

F.4: As in RC1, recommend that you minimize the number of different vertical variables, even if you end up writing ($h\_i$-$h\_j$) in text rather than derived 'z' quantities. The fundamental measured values are antenna height above ellipsoid/geoid, and antenna height above snow/firn surface, right? This figure would also be a good place to define distinctions between snow and firn, and between firn and ice.

F.5: Minor point, but commit to all caps everywhere, including figure legends, for site names etc. (e.g., "pig1"=>"PIG1" etc)

F.8 and F.9: switched order?

F.9: stack panels according to order of introduction in text. (Or modify text, if you think the figure panel order is more logical.)

F.10: mark "along-flow" and "transverse" directions on at least one panel.

---

## Author Comment (AC3) · 2 Jul 2017

[revised manuscript text omitted]

.

Observed surface Dh

| Page 24: [17] Deleted Cells | David Shean | 7/1/17 5:59:00 PM |
|---|---|---|

Deleted Cells

| Page 24: [18] Deleted Cells | David Shean | 7/1/17 5:59:00 PM |
|---|---|---|

Deleted Cells

| Page 24: [19] Deleted Cells | David Shean | 7/1/17 5:59:00 PM |
|---|---|---|

Deleted Cells

| Page 24: [20] Deleted Cells | David Shean | 7/1/17 5:59:00 PM |
|---|---|---|

Deleted Cells

| Page 24: [21] Deleted Cells | David Shean | 7/1/17 5:59:00 PM |
|---|---|---|

Deleted Cells

| Page 24: [22] Deleted Cells | David Shean | 7/1/17 5:59:00 PM |
|---|---|---|

Deleted Cells

| Page 24: [23] Deleted | David Shean | 7/1/17 5:59:00 PM |
|---|---|---|

| Page 24: [23] Deleted | David Shean | 7/1/17 5:59:00 PM |
|---|---|---|

| Page 24: [24] Deleted Cells | David Shean | 7/1/17 5:59:00 PM |
|---|---|---|

Deleted Cells

| Page 24: [25] Deleted Cells | David Shean | 7/1/17 5:59:00 PM |
|---|---|---|

Deleted Cells

| Page 24: [26] Deleted | David Shean | 7/1/17 5:59:00 PM |
|---|---|---|

| Page 24: [26] Deleted | David Shean | 7/1/17 5:59:00 PM |
|---|---|---|

| Page 24: [27] Deleted | David Shean | 7/1/17 5:59:00 PM |
|---|---|---|

| Page 24: [27] Deleted | David Shean | 7/1/17 5:59:00 PM |
|---|---|---|

| Page 24: [28] Deleted | David Shean | 7/1/17 5:59:00 PM |
|---|---|---|

| Page 24: [28] Deleted | David Shean | 7/1/17 5:59:00 PM |
|---|---|---|

| Page 24: [29] Deleted | David Shean | 7/1/17 5:59:00 PM |
|---|---|---|

| Page 24: [29] Deleted | David Shean | 7/1/17 5:59:00 PM |
|---|---|---|

| Page 24: [30] Deleted | David Shean | 7/1/17 5:59:00 PM |
|---|---|---|

| Page 24: [30] Deleted | David Shean | 7/1/17 5:59:00 PM |
|---|---|---|

| Page 24: [31] Deleted Cells | David Shean | 7/1/17 5:59:00 PM |
|---|---|---|

Deleted Cells

| Page 24: [32] Deleted Cells | David Shean | 7/1/17 5:59:00 PM |
|---|---|---|

Deleted Cells

| Page 24: [33] Deleted Cells | David Shean | 7/1/17 5:59:00 PM |
|---|---|---|

Deleted Cells

| Page 24: [34] Deleted Cells | David Shean | 7/1/17 5:59:00 PM |
|---|---|---|

Deleted Cells

| Page 24: [35] Deleted | David Shean | 7/1/17 5:59:00 PM |
|---|---|---|

| Page 24: [35] Deleted | David Shean | 7/1/17 5:59:00 PM |
|---|---|---|

| Page 24: [36] Deleted | David Shean | 7/1/17 5:59:00 PM |
|---|---|---|

 and Dh/Dt results.

| Page 24: [36] Deleted | David Shean | 7/1/17 5:59:00 PM |
|---|---|---|

 and Dh/Dt results.

| Page 24: [36] Deleted | David Shean | 7/1/17 5:59:00 PM |
|---|---|---|

 and Dh/Dt results.

| Page 24: [36] Deleted | David Shean | 7/1/17 5:59:00 PM |
|---|---|---|

and Dh/Dt results.

| Page 24: [36] Deleted | David Shean | 7/1/17 5:59:00 PM |
|---|---|---|

and Dh/Dt results.

| Page 24: [36] Deleted | David Shean | 7/1/17 5:59:00 PM |
|---|---|---|

and Dh/Dt results.

| Page 24: [36] Deleted | David Shean | 7/1/17 5:59:00 PM |
|---|---|---|

and Dh/Dt results.

| Page 24: [37] Deleted | David Shean | 7/1/17 5:59:00 PM |
|---|---|---|

for both the surface Dh/Dt and corrected pole-base $Dh_{pb}$/Dt (

| Page 24: [37] Deleted | David Shean | 7/1/17 5:59:00 PM |
|---|---|---|

for both the surface Dh/Dt and corrected pole-base $Dh_{pb}$/Dt (

| Page 24: [37] Deleted | David Shean | 7/1/17 5:59:00 PM |
|---|---|---|

for both the surface Dh/Dt and corrected pole-base $Dh_{pb}$/Dt (

| Page 24: [37] Deleted | David Shean | 7/1/17 5:59:00 PM |
|---|---|---|

for both the surface Dh/Dt and corrected pole-base $Dh_{pb}$/Dt (

| Page 29: [38] Deleted | David Shean | 7/1/17 5:59:00 PM |
|---|---|---|

: A) Observed cumulative displacement between SOW1 and other 2012–2014 stations. Legend lists initial distances. B) Observed cumulative strain, with best fit strain rate listed in legend. C) Smoothed residuals from linear fit, highlighting subtle variations and inflections in strain rates. D) Daily GPS velocity. Note timing of abrupt absolute velocity changes observed at all sites and inflection points in observed strain across the array.

[Figure]

| Page 31: [39] Deleted | David Shean | 7/1/17 5:59:00 PM |

Plots of observed and modelled values for GPS sites (see Figure 4 for reference). A) Reflector height ($z_{rh}$) relative to initial surface ($h_0$). Positive values indicate surface height increase relative to GPS antenna. Legend indicates initial antenna height above surface. Note ~20–30 cm surface decrease from December 2012 to January 2013. B

| Page 31: [40] Deleted | David Shean | 7/1/17 5:59:00 PM |

$h_{pb}$

| Page 31: [41] Deleted | David Shean | 7/1/17 5:59:00 PM |

fit, which can be directly compared with *Dh/Dt* fits in D (see Table 1). Note reduced

---

## Author Response (AR1)

Review of:

**In-situ GPS records of surface mass balance and ocean-induced basal melt for Pine Island Glacier, Antarctica**

*By David E. Shean et al.*

This paper discusses time series measurements of surface elevation change made using an array of GPS receivers set out on the floating tongue of Pine Island Glacier. These and other data are used to quantify the processes contributing to changes in ice thickness, especially the surface and basal mass balance. The analyses are described in some detail and the results are interesting. The work is therefore certainly publishable, but my feeling is that more work should be devoted to the presentation to make the results more accessible.

We thank the reviewer for detailed comments and suggestions.

At present there is no clear logical flow from definition of the problem, through observation and analysis of observation and ancillary data, to conclusions. The discussion jumps between observations, processes, models, supporting data, and it is difficult to keep track of what has actually been observed and what the ultimate goal of the analyses is.

The original organization attempted to isolate data/methods, observations, interpretations, and speculation.

The last two paragraphs of the introduction define the problem (lack of long-term BMB records and validation for SMB/firn models), the approach, and the overall goals of the paper.

We modified the text to clarify the motivation and reorganized to address many of the issues raised by the reviewer.

A bewildering range of surface height variables are introduced, many of which are not independent but simply derivatives of the others, and this really adds to the (unnecessary) complexity of the presentation.

We made major revisions to simplify the coordinate system and number of variables.

The problems are immediately apparent in that abstract. We are told that time series of "antenna height, surface elevation and Lagrangian elevation change (Dh/Dt)" are to be presented. But without any definitions the reader is left to guess what these variables actually are and how they

are distinct.

We modified the abstract to improve clarity.

Reading the paper, it seems that "height" refers to measurements relative to the instantaneous snow surface, while "elevation" refers to measurements relative to the fixed geoid. However, "elevations" are confusingly denoted by the symbol "h", and the terminology is not used entirely consistently.

We spent considerable time debating the convention to use for this paper before submission. In the end, we elected to use the traditional glaciology convention for the submission, where h is surface elevation (for both snow and ice) above some datum (WGS84 ellipsoid or geoid).

We agree that surface elevation (z), height (h) and thickness (H) would be easier to understand, and modified the notation accordingly.

The next sentence in the abstract says that "The antenna height time series show a surface elevation increase". If the increase is relative to the geoid, the ice must the thickening, right? Apparently not. Later on we are told that "observed Dh/Dt" is negative, implying thinning. This is probably the worst example of inconsistent terminology, but the discussion could be simplified throughout by reducing the number of variables to just the independent ones and a couple of critical derived ones.

The increase was relative to the initial surface, not the geoid. Previous work with GPS interferometric reflectometry (Larson et al., 2015) introduced the "reflector height" term. We agree that this is confusing, so we redefine "antenna height" as "antenna-surface distance" and simplify as mentioned above.

To add to the current confusion, the schematic defining all the variables (figure 4) actually shows Dh/Dt as positive.

This figure has been simplified, and we changed to show Dh/Dt as negative. The original intention was to show that antenna and surface elevation change are evolving independently, but we agree that the presentation was confusing.

The most fundamental variable measured by the GPS is the antenna elevation. A subsidiary measurement, and a really nice addition, is the height of the antenna above the snow surface. To avoid potential confusion between elevation and height (especially if the authors are not going to be entirely rigorous about their usage), I would recommend using an entirely different name for

antenna height, perhaps "ground clearance" or "exposed pole length".

We appreciate what the reviewer is suggesting here, but do not feel either of the terms suggested are appropriate. The offset is not just exposed pole length, as it is measured to the phase center of the GPS antenna. We define as "antenna-surface distance"

Apart from these two variables, nothing is directly measured by the GPS. Everything else is a derived quantity, so why not just stick to these, unless it is absolutely essential to introduce something new. One derived variable that is worthwhile having is the snow surface elevation, since that is what the satellite stereo imagery (introduced later as an ancillary dataset) measures, but that is just the antenna elevation minus the exposed pole length. The "pole base elevation" is a useful reference level, but is just the antenna elevation, minus the fixed pole length, unless the pole angle relative to the vertical changed with time (a possibility that does not seem to have been considered). I don't see the point of introducing the "reflector height" variable which is just the initial "antenna height" (or ground clearance/exposed pole length) minus its initial value. So trends in "antenna height" and "reflector height" are identical, apart from the sign. Arguably the "pole base depth" below the surface (or "buried pole length") is useful in that compaction rates above this level and below it ultimately need to be considered separately.

We agree, and focus our formulation using the two observables. We chose to preserve a few other quantities (surface elevation) and attempted to better define the "surface elevation relative to tracer for initial surface elevation" which is not a simple inversion of antenna-surface distance.

By the way, given that one pole fell over completely, can you be sure that your measurements are free of contamination from changes in the pole angle? Were there repeat measurements of pole angle?

None of the original authors responsible for data analysis and manuscript preparation were present during installation and servicing of these GPS stations, and there is very limited documentation available due to the rushed nature of these field campaigns. We made several requests to original field team members for additional information about the installation and servicing, but did not receive definitive responses on several items. Limited photographs show that the poles appear vertical during what may be servicing visits. Personal communication with M. Truffer and communication with newly added co-authors D. Holland and M. Bushuk lead us to believe that the poles did not tilt.

If there was significant tilting of the pole, we should see much larger spreads in the reflectometry measurements from GPS signals in different quadrants. We added some text to address the issue

of pole tilting in Section 5.2 (now, Pole-base settling/tilting), considering errors introduced by 10 and 20° tilts, which are negligible for vertical elevation change.

You also make the assumption that the pole base is the point that is fixed in the firn. That seems reasonable to me, but did you put an insulating stopper in the end of the pole? That would make your assumption more justifiable.

This information was not available in archived documentation, but D. Holland suggested that there were indeed insulating stoppers at the pole base.

So for the GPS time series there are two measured quantities (antenna elevation and exposed pole length) and two derived quantities (surface elevation and buried pole length). I don't really see why you need any more. These can then be used to derive the surface and basal mass balance. The most consistent way of doing this would be to define a local vertical coordinate that is zero at the base of the pole and positive upwards.

We disagree with the reviewer about the choice of the pole base as the origin. The antenna phase center is the most accurate position we have, and we choose to use this as our origin.

Then taking your equation (3) and substituting:

where $ia$ is the ice-atmosphere interface, $io$ the ice-ocean interface and $d_a^{+/-}$ is the thickness of air above and below the origin, you get two separate equations for the surface and basal mass balance:

because the reference surface is assumed to be a material surface. Now, some of these terms you have implicitly ignored (probably perfectly valid, but it would have been helpful to see the process and justification of the simplifications), and of the others, some come from your GPS measurements and some from ancillary data and assumptions.

We appreciate the reviewer's formulation. We reworked our original derivation for basal mass balance and further explained our assumptions.

The divergence term comes from your measurements of the rate of change of the distances between GPS stations. Or does it? The methods section describes how you derive absolute positions and motion relative to fixed reference stations. But the absolute motion is unimportant here as your calculations are in a Lagrangian framework. Couldn't the inter-stake distances be derived more simply and accurately by differential positioning of one station relative to the

other? No fixed base station is needed then.

The reviewer misunderstood our methods here.  Following standard differential GPS processing methodology, we used a fixed base station to correct all GPS records with the goal of obtaining highest possible absolute accuracy at all sites. The differential correction applied at all 5 sites was effectively the same.  The relative accuracy for inter-site distances should be ~1 cm, so it is unnecessary to perform differential corrections within the network.

We then calculated inter-site distances within the 5-site network. Figure 6 shows inter-site distances calculated relative to SOW1, not the fixed base station.  We performed similar calculations using each station as a reference, but only included the figure with SOW1 as reference in the paper.

And given that you have all the inter-station distances as a function of time, why don't you make a direct estimate of the divergence from those data?

We did – that is how we estimated longitudinal/transverse strain rates within the array (included in original submission text), which we multiply by thickness estimate to obtain flux divergence and expected elevation change due to flux divergence (which is negligible).

Our preliminary analysis produced the following figure, which shows the direct inter-station distances as a function of time for the full array. Figure 6 from the initial submission included the first column (refer to original caption), as we felt the figure was too busy.

[Figure]

Actually, since you have five stakes, and you only need three to do the calculation, you can do it with multiple stake combinations and get some idea about the spatial variability. Why haven't you done that? I found your ad hoc calculations based on the assumption that some stakes were oriented along flow, and that along-flow extension should dominate, to be rather unsatisfactory.

We did make a direct estimate of divergence, as shown in Figure 6 and described in Section 3.2 of the original submission.

Our assumption about the orientation of the sites was not ad hoc. The sites were installed in this along-flow and cross-flow orientation based on known velocity vectors for PIG. Our initial calculations between each pair of stations shows that magnitude of the strain rates is negligible, so we did not pursue a more rigorous analysis.

Regardless, we reworked our strain rate estimates for 8 different triangular sections within array. The figure and text were updated accordingly. While this analysis may be more rigorous, we actually lose some of the temporal resolution from the direct site to site displacement time series in Figure 6 of the original submission, which provides useful information about the timing of changes in velocity and strain rates.

The air thickness terms come from the firn modelling, which needs a much clearer discussion of the inputs and outputs. I think it is driven by the "SMB and Temperature data" discussed in section 2.4. In which case, why is that section not a part of a larger section that discusses the firn model?

This is not the case. Our estimate for air thickness comes from an analysis of all available airborne radar two-way travel time and contemporaneous surface elevation from airborne LiDAR (see Shean, 2016). For this paper, we considered the temporal evolution of firn air content from FDM to show that it does not vary much during the ~2-year time periods, but did not use the absolute firn air content estimates from the model.

As it stands, it is presented to the reader as a standalone estimate of SMB, but that is one of the quantities promised in the title from the GPS records. The point being, I assume, that you want regional estimates to drive the firn model, distinct from the point estimates that are the result promised in the title? How sensitive are your firn model results to these inputs?

We include references to several existing publications that describe the firn model and input/output data in detail. These details are beyond the scope of this paper. We did not use our GPS-derived results to drive the firn model.

Basing your temperature on a measurement at Evans Knoll and an assumed lapse rate will introduce a warm bias. It is well known that a stable inversion layer can form over the flat topography of the ice shelves, so surface temperatures can be lower than those on surrounding higher ground. I doubt that ERA-Interim captures this. Could you not use remote sensing data?

We included the Evans Knoll temperatures and the ERA-Interim temperatures in the paper for reference, as they provided the best available T information at the time. The NYU AWS data from the PIG drilling camp were not publicly available.

We have since added David Holland and Mitch Bushuk to the author list, as they provided access to the NYU AWS data. Unfortunately, the NYU AWS moved locations from 2008-2010, 2011-2012, and 2013-2015. It was co-located with the GPS network from Jan 2013 to Jan 2014. We use the 2013-2015 period to compute a more appropriate scaling of the longer 2011-2015 Evans Knoll AWS data and the ERA-Interim data. We are most interested in relative T records to provide context for the warm periods where we document large elevation change.

We acknowledge that there are a number of issues with the AWS records, but it is a minor component of the paper. A separate, more detailed study of the AWS data is warranted.

What remote sensing data does the reviewer have in mind? It is unlikely that any remote sensing data with sufficient resolution would provide the necessary temporal resolution for this comparison.

The outputs from the firn model are never clearly presented. A vertical velocity associated with compaction is quoted in a number of places, but the separation of compaction into the components that occur above and below the pole base (as in the above equations is never made clear). These are key components of the results, and I would have expected to see some graphical presentation of them. The surface height from the firn model is shown, but this doesn't tell the reader what was actually used in the calculations of surface and basal mass balance.

We elected not to include the firn model details in the original submission, as these are described in existing literature. We include some figures here for clarification, and can consider including as supplemental figures with the manuscript if necessary.

The simulated IMAU-FDM surface elevation profiles do include the compaction above and below the pole base.

[Figure]

Figure 1: PIG shelf grid cell IMAU-FDM surface velocity components (for detailed description, see Ligtenberg et al., 2011). $v_{ice}$ is downward velocity across the firn-ice transition, which is set to the long-term ice-equivalent SMB. Also note that $v_{bouy}$ term counters positive velocity due to accumulation ($v_{acc}$), as this grid cell is floating. Expected IMAU-FDM surface elevation is computed by integrating $v_{total}$.

[Figure]

Figure 2: Expected elevation of tracers for the original surface ($\widetilde{z_{surf0}}$)) and pole base (assuming initial depth pole base depth of 1.0 m), relative to initial surface elevation, from 2012-2014.

[Figure]

Figure 3: Expected surface ($\widetilde{z_{surf}}$, **dashed lines**) and pole base ($\widetilde{z_{polebase}}$, **solid lines**) elevation. Note that pole base motion appears similar for the observed range of pole base depth below the surface (~0.5–1.0 m).

The $z_{ia}$ term is simply the (assumed fixed) pole length minus the (measured) exposed pole length. The $z_{io}$ term is a bit more complex, since it must come from the measurements of antenna elevation and an assumption of isostatic equilibrium:

where $h_0$ is the elevation of the reference surface above the geoid, that is the measured antenna elevation minus the fixed pole length. This leads to:

which is now clearly in terms of the two directly measured quantities and two outputs from the firn model. Now the melt rate can be derived from the following equation:

.

In the above notation $h_0$ is the same as the authors' "pole base elevation", so I think a couple of the terms in this equation are the same as those in their equation (7), but I am not sure about the rest. I did not really follow all the assumptions made in the derivation of (7) and I would argue that the above is clearer in its relationship to the directly measured antenna elevation, the firn model output and the surface accumulation rate (itself derived above from a directly measured quantity and firn model output).

We appreciate the reviewer's take on the problem. We modified our derivation to accommodate several of these issues, but elected not to use a pole base origin.

Of course, this equation and the authors' version relies on the assumption of isostatic equilibrium. Curiously the validity of this assumption is never discussed.

We added a new section on hydrostatic equilibrium to the discussion section. More discussion on the validity of this assumption is included in a forthcoming companion paper on the DEM analysis for the PIG shelf.

However, the quoted thicknesses measured from radar differ from those calculated from surface elevation, so it is clear that the assumption does not hold.

We strongly disagree with the reviewer's opinion that this difference means the assumption does not hold.

The Stanton et al. (2013) radar processing did not include corrections for variable radar velocity in snow, firn, and ice.  This likely leads to an overestimate of the radar thickness in the Stanton et al (2013) results.  We also do not know definitive information about the position and acquisition dates of the radar profile relative to the GPS array, as this is not archived/documented. Repeated inquiries about relative locations of the PIG field data collection did not provide detailed answers.

We also do not have well-constrained estimate of local firn air content, so we use our shelf-wide estimate as described earlier.  On some level, this firn air content uncertainty is less important, as it falls out of the elevation change term, and we know that the velocity divergence term is small, due to limited strain rates.

Herein lies the major weakness of the paper. It is well-known that over length-scales comparable with the ice thickness, vertical shear stresses can partially support the ice.

We also disagree that the assumption of hydrostatic equilibrium is a "major weakness."  This is addressed in the new discussion section on floatation.

The GPS network is ~2 km across, which is >4-5x the local ice thickness (~350-500 m).  The nearby channel/keel widths are ~1-2 km across, so still >2-3x the local ice thickness.  We argue that for the observed length scales and thicknesses, the hydrostatic assumption is perfectly reasonable.

This has been shown to be the case on similar channel-like features on a number of ice shelves (McGrath et al., 2012, Ann. Glaciol., 53, 10-18; Jenkins et al., 2006, J. Glaciol., 52, 325-346), including on Pine Island (Vaughan et al., 2012, J. Geophys. Res., 117, F03012).

Experiments with a high-resolution ice-flow model show that wider channels tend to be near equilibrium, with bridging stresses becoming more important for narrow channels (Drews, 2015).

It is also unclear whether the studies cited by the reviewer used appropriate sub-km-scale snow/firn thickness variations when processing radar data or removing firn air content when calculating deviation from hydrostatic.

The key point is that while the ice is not freely floating (as it is not on the scale of the stake network discussed here) the regions of thinner ice will be sinking as the ice deforms under the vertical shear stress.

Many of the papers suggested by the reviewer involve processes that are much more important closer to the grounding line. The section of the PIG shelf with the GPS receivers is relatively thin (~450 m) and based on its distance from the grounding line (~40-50 km in 2012) and known ice shelf velocities (~3.8-4.0 km/yr), this section had at least 10-12 years to equilibrate after crossing the grounding line before our study period starts. We argue that the magnitude of any vertical "sinking" of thinner ice should be significantly dampened by 2012.

So if you set up a GPS station in a channel you will see a surface lowering even if the ice thickness remains constant. Such a process could introduce a significant bias to the estimates of basal mass balance that does not appear to have been considered by the authors.

Again, this lowering rate will be highest near the grounding line, and should decrease over time.

The magnitude of this lowering is also highly dependent on the actual thickness difference between the channel and the adjacent keels. Near the GPS receivers, the observed surface elevation difference between a trough floor and an adjacent ridge crest is typically less than 10 m or <15%. It is unlikely that this process would result in 3-5 m/yr of surface lowering, especially for the timescales involved.

We disagree with the reviewer that this effect introduces a significant bias in our basal mass balance estimates.

It might, for example, explain why the surface elevation changes are rather steady despite the change in ocean conditions that have been inferred to have changed the melt rates.

We strongly disagree with the suggestion that the relaxation would be large enough to dominate the observed surface elevation change. We arrive at the same conclusion regarding ocean sensitivity when using Dh/Dt from high-resolution DEMs over the entire shelf, for multiple time periods and spatial scales.

Overall, my recommendation would be to restructure the paper along the following lines:

Introduction: A general introduction to the region and the problem to be addressed. Don't discuss your GPS sites in detail yet.

Method: Describe the principle of how measurements of the elevation and exposed pole length can be used to calculate the surface and basal mass balance. Introduce the reference frame and variables and show what additional information you need.

GPS data: Describe the set-up of the GPS sites, the data collected and the processing to get to the key results of antenna elevation and exposed pole length and inter-pole distances, and the uncertainties in all these numbers.

Divergence: Describe the calculations and show the horizontal variability. Discuss the uncertainties.

The horizontal variability is a result. It seems the reviewer is suggesting that this be mixed with description of the methodology?

Firn compaction: Describe the model and the necessary input data. Show the outputs that you actually use in your calculations.

Results: Calculate surface and basal mass balance and uncertainties. Compare with other studies.

Discussion: Discuss all the other potential sources of error. Don't forget the impact of adjustment towards isostatic equilibrium. Not only is it a source of elevation change that is not related to thickness change, but it gives rise to a flow divergence that changes with depth, so your measurements of surface divergence are not representative of the depth-mean.

Conclusions: Can you say for sure that the melt rates were steady, given the uncertainties and potential biases? If it was, how do you explain that observation?

Finally, some more minor points:

I would recommend making it clearer in the title that the measurements are on the floating part of Pine Island Glacier and that the surface and basal mass balance estimates are derived from GPS measurements. The wording at present suggests that the results are directly measured by GPS.

The PIG1 site is not on the floating portion of the PIG shelf.  Regardless, we changed the title, as most of the paper focuses on the shelf.

Some of the figures could be improved. In particular, it would be much easier if the sites were labelled in Figure 2. The colour code is really hard to use. Again, label them on Figure 10.

The color code is consistent throughout the text, and we feel it is useful for connecting the maps with the time series plots.  We chose not add labels to Figure 2 as we did not want to obscure local surface topography near the GPS sites.  We added labels to Figure 10.

Why did you use a tide model to remove the tides from your records? Why not do a harmonic analysis of the elevation data themselves? Your records are long enough to make reliable estimates of more tidal components than are given in the CATS output, aren't they?

The tide model (which is excellent) was used to remove the expected tidal fluctuations. A subsequent frequency filter was used to remove residual components and tide model errors with period <1.5 day. The CATS2008A tide model showed excellent agreement with the observed GPS records. Using a harmonic analysis would not change the results in a significant way.

In summary, while this is a long review, there are some really nice results contained in this paper. I also think the detailed discussion of the data and analyses could be a real strength, if it were presented in a more readily accessible way. The current presentation left me a little confused as to what the bottom line really is. The title promises estimates of both surface and basal mass balance, but the calculations are dependent on the results of a firn model that needs surface mass balance as an input. It is never made clear to what extent the author's regard their calculations of surface accumulation as independent results, or if the discussion is merely a demonstration that measured changes in exposed pole length are consistent with, rather than an addition to, prior knowledge.

The reviewer is correct, that we are using modeled SMB to drive a firn model, and then comparing with observations. We improved the text to clarify, and then emphasize the main results involving basal melt rates. The RACMO SMB is used to provide necessary first-order corrections, and the GPS results provide an improved record of ~daily accumulation. We demonstrate that there is more to be done with the GPS-derived accumulation records, but this is better suited for a follow-on study.

Likewise, with the basal mass balance. Do those results represent an addition to our knowledge of melting at the base of Pine Island, or are they more a demonstration of the limitations of using surface elevation data to infer melt rates at these spatial scales?

Our results present new information about basal melting, and demonstrate that this information can be derived from GPS records.

The 2-year GPS records provide important constraints for a period when ocean temperatures were variable. These records also provide important cal/val for melt rates derived from the high-res DEMs.

Existing knowledge about PIG melt rates come from a single site described in the Stanton et al

(2013) paper for a ~30 day period, with limited information about actual location.

**L. Padman (Referee)**

padman@esr.org

SUMMARY

This paper describes a GPS-based technique to make estimates of basal mass balance (BMB) and surface mass balance estimates for Pine Island Glacier ice shelf (PIGIS). The inclusion of GPS-derived snow surface height relative to the GPS antenna is a particularly interesting addition to standard ice-shelf GPS analyses, and the results are tied in well to other utilized datasets including the stereo-imagery of the DEM and the surface snow/firn state from a RACMO-based firn density model (FDM).

I have the advantage of being late in my review, so I don't need to make the detailed review comments in RC1. However, I agree with all of these. In particular:

(a) It took me a long time to get the actual vertical coordinate system sorted out in my head. The "surface relative to the receiver" is a great addition, but it means finding clearer symbols, and explaining the coordinate system more carefully, early. (e.g., Already by line 15 in the Abstract, I was confused by the statement that "The antenna height time series shows a surface elevation *increase*" on an ice shelf that is thinning rapidly.) RC1 suggested fewer symbols. I agree with this, and suggest using relative height values as close to fundamental measurements as possible (the 'h' variables in Fig. 4), rather than derived difference quantities ('z' variables) even if it means adding text volume. Then also use longer but more descriptive subscripts. So, e.g., instead of $z_a$, use ($h_{antenna} - h_{snow\_sfc}$).

We modified the variables and coordinate system as described earlier and included more descriptive subscripts.

(b) The issue of whether the hydrostatic assumption is really appropriate in the presence of small-scale ice thickness variability needs to be better addressed. Maybe this has something to do with the obscure (to me) "diffusion of local topography"? But somehow convince us that hydrostatic is okay.

Addressed in earlier responses to RC1 and in Discussion section.  Reworded the "diffusion of local topography"

(c) I agree with RC1 on the general layout of the revised paper. Revisions responding to RC1 cautionary comments about data analyses (e.g., maybe other poles also lean?) run the risk of weakening your conclusions specifically about BMB. However, I don't think that matters too much, as your paper will still describe an interesting approach to GPS analysis that will likely lead to much improved GPS installation practices and data value from future GPS sites.

We disagree with the suggestion that our revisions weaken conclusions about BMB, and back this up with additional discussion in the revised paper.  We appreciate the reviewer's comment

about applicability to other GPS sites.

Specific comments are divided into "Major", where I'd like to know what you did in response, and "Minor", which don't require documentation in revision. Numbers refer to original page.line.

MAJOR COMMENTS

Title is not right. First, what "*not* in-situ" method is available for GPS? Second, there are other important data sets and model output sets used here.

Removed "in-situ" from title and modified as "GPS-derived estimates".

1.15: This is when it first became clear I didn't understand the vertical coordinate system; see SUMMARY comments and RC1.

Modified as described earlier.

3.7-8: The Jacobs et al. (2011) paper compares just one cruise in 1994 with another in 2009. This doesn't seem like robust evidence for a causal link between the "increased ocean heat content" and accelerated mass loss. It probably is, although glacier dy- namics must figure into this, so maybe put an "attributed to" in there.

Good point.  Modified text accordingly.

3.9-10: The 40-50 GT/a is *net* mass loss *from the grounded glacier*, right? Other- wise these numbers don't make sense given 130 Gt/yr discharge. Need to be specific.

Added "net" for clarification.

6.13-25: What does a bias in the atmospheric model do to the results from a firn density model? And is the bias even taken into account when the FDM is run? Since RACMO2.3 is run from ERA-Interim (I think), then presumably RACMO2.3 has similar biases that need to be corrected for before running the FDM.

A bias or unknown long-term trend in the modelled atmospheric conditions can indeed have an effect on the IMAU-FDM results. For the IMAU-FDM initialisation, it is assumed that the climate prior to the simulation period (i.e. pre-1979) has been similar to the 1979-2015 average climate and to accommodate any unknown trends before 1979 a sensitivity analysis was performed (see SOM of (Pritchard et al., 2012)). It is found that for high accumulation and/or high surface melt regions the uncertainty in modelled surface elevation can be as high as 20% (2-sigma confidence interval). Since the PIG shelf is not an extreme accumulation region (< 1000 mm w.e.) and experiences moderate surface melt, the modelled surface elevation uncertainty is ~10%. Unfortunately, the uncertainty analysis has not been performed on the individual vertical velocity components (e.g. v_fc). As it is unlikely that the uncertainty/bias in an individual components is much larger than the total uncertainty, we take a conservative estimate for the uncertainty in v_fc of 10% (or ~0.1 m yr-1).

We are aware that a possible bias in ERA-Interim is transferred into RACMO2 and therefore in

the IMAU-FDM results. However, RACMO2 results are extensively evaluated by numerous independent methods and the expected biases are within the above-described uncertainty estimates.

Finally, the absolute bias is less important for the relative evolution during the 2-year periods under consideration in this paper.

9.8: What is "diffusion of local topography"?

Changed to "local deformation" and added more discussion of bridging stresses and relaxation.

9.34: "reflector height" is a confusing term. See extended discussion of coordinate and naming issues in RC1.

This was the term used in previous literature, and we agree it is confusing. Changed to "antenna-surface distance"

10.35: Identify, on Fig. 10, the "flow" and "transverse" directions.

We added a flow direction arrow to Figure 2, which should be sufficient.

MINOR COMMENTS

3.9: either "with annual discharge of ~130 Gt" or "with discharge of ~130 Gt/year".

Changed.

5.2-3: I agree with RC1 that it doesn't make sense to detide with a tide model when you have tide-resolving GPS. It's nice to see that CATS works well, so keep the comparison, but detiding with a tide model only makes sense for short or gappy GPS records. A better approach would be using T-Tide (in matlab) or the equivalent Foreman (1977) FORTRAN code.

We appreciate the suggestion, and will consider for future analyses, but feel that our approach is sufficient given the goals of the study and our focus long-term change. We removed the tidal signals using CATS, and then applied a 1.5-day low-pass filter, which removed residual high-frequency errors and provided smoothed time series for further analysis. Rerunning with the suggested approaches will not significantly change our results.

5.25: Does "firn" include "snow"? Fig. 4 suggests that snow and firn are regarded distinctly. If so, the interface you are getting the relative surface elevation from is not necessarily "firn-air"

This page and line number do not include any mention of firn or snow.

Fig 4 was meant to be illustrative, with "snow" representing new snow accumulation since the installation at time t0. As soon as the "snow" is emplaced, it is included in the dynamic firn model, so, yes, "firn" includes "snow"

6.21: 917 kg/m^3 is solid ice, right? So it's a bad criterion for the firn-ice transition.

Yes, solid ice is assumed to have a density of 917 kg m-3. Therefore, in the IMAU-FDM we take the depth at which this density is reached as the boundary between the firn layer (all densities < 917 kg m-3) and ice column (density equals 917 kg m-3). With the IMAU-FDM, we try simulate all processes in the firn layer in order to be able to divide elevation changes between changes occurring in the firn or ice column. Our opinion is that the firn-ice transition is best defined as the depth at which the density reaches 917 kg m-3.

page 7 and elsewhere: Once the vertical coordinate symbol set is finalized, make sure all occurrences are formatted the same, e.g., \it{H}, not H.

Checked all formatting for consistency.

7.25: 5 kg/m^3 seems like a high error for water density, and probably also high for solid ice, although there is might be used as a fudge factor for firn density profile.

This was a conservative estimate, but we agree it is probably too high. Changed to 1 kg/m3, which slightly reduces basal melt rate uncertainty estimates.

8.14: "30-40 m/yr" doesn't seem "subtle" to me.

Good point. Removed "subtle"

8.17: velocity units on Fig. 5 are m/y, so don't use cm/day in text.

This is describing velocity change over timescales of days, so units of m/yr are not appropriate. Changed to "m/day" and reference to Fig 6, which does include units of m/day.

9.22: (a) first call to Figure 9 is to Fig. 9C. Organize panels in the order that they come up. (b) Figure 9 called out before Figure 8?

We reorganized panels and references in text should be consistent.

10.5: either "from May to August" or "for the period MayâA ˇTˇAugust"

Replaced "–" with "to"

10.18 and 16.12: What is "NMAD" ?

This is defined as normalized median absolute deviation on Page 6 when describing AWS data.

10.25-27: Not sure what you mean by "aggregated" here.

Changed to "Composite products with time interval between 0.5–2.5 years were generated"

This process is described in greater detail in forthcoming paper focused on the DEMs, rather than GPS.

10.34: delete "(Figure 10)"; sentence already starts "Figure 10 shows"

Deleted.

11.5: In text you quote mwe/month but in the figure you send us to (9F), SMB is given as mwe/day, so I can't directly compare them.

We chose to scale the monthly SMB to mwe/day rather than scaling the daily SMB to mwe/month, which would lead to a confusing range of large values.

12.2: delete "the" in "The results in the Section 3"

Done

14.33: repeat "bottom" in "bottom altimeter" here. I lost the thread of what altimetry you were talking about. Alternatively, reorganize this to tell us all about the altimeter then compare its results with your estimates.

Repeated "bottom" which we feel is sufficient, as we are not directly showing other altimetry data for this paper.

15.11-14: This text could be tightened up.

Modified the text to improve clarity.

FIGURES

See RC1 for some other figure comments.

F.1: Add a bold "North" (N) arrow since occasionally you refer to directions and it is hard to get oriented with this map.

Added.

F.2A: again, north arrow would be good, and a flow-direction arrow would help. (Better than just having it in the caption.)

Added a flow direction arrow, but did not add N arrow, as this figure is already cluttered and the location is clearly shown in figure 1.

F.4: As in RC1, recommend that you minimize the number of different vertical vari- ables, even if you end up writing (h_i-h_j) in text rather than derived 'z' quantities. The fundamental measured values are antenna height above ellipsoid/geoid, and antenna height above snow/firn surface, right? This figure would also be a good place to define distinctions between snow and firn, and between firn and ice.

Simplified (see response to RC1).  It is unclear about what "define distinctions" means.  We have labels and lines for the air/snow and snow/firn transitions.  We do not feel it is necessary to include the firn/ice transition, as this is much deeper in the column.

F.5: Minor point, but commit to all caps everywhere, including figure legends, for site names etc. (e.g., "pig1"=>"PIG1" etc)

Done.

F.8 and F.9: switched order?

Reviewed and ensured that text and figures are in order.

F.9: stack panels according to order of introduction in text. (Or modify text, if you think the figure panel order is more logical.)

Done.

F.10: mark "along-flow" and "transverse" directions on at least one panel.

We added a flow direction label to Figure 2 as suggested, which eliminates the need to add labels to figure 10. Also, we are confident that the reader can infer relative motion of the GPS sites between A and C in Figure 10.

References:

Drews, R.: Evolution of ice-shelf channels in Antarctic ice shelves, The Cryosphere, 9(3), 1169–1181, doi:10.5194/tc-9-1169-2015, 2015.

Jacobs, S. S., Jenkins, A., Giulivi, C. F. and Dutrieux, P.: Stronger ocean circulation and increased melting under Pine Island Glacier ice shelf, Nat. Geosci., 4(8), 519–523, doi:10.1038/ngeo1188, 2011.

Joughin, I., Smith, B. E. and Holland, D. M.: Sensitivity of 21st century sea level to ocean-induced thinning of Pine Island Glacier, Antarctica, Geophys. Res. Lett., 37(20), n/a-n/a, doi:10.1029/2010GL044819, 2010.

Larson, K. M., Wahr, J. and Kuipers Munneke, P.: Constraints on snow accumulation and firn density in Greenland using GPS receivers, J. Glaciol., 61(225), 101–114, doi:10.3189/2015JoG14J130, 2015.

Ligtenberg, S. R. M., Helsen, M. M. and van den Broeke, M. R.: An improved semi-empirical model for the densification of Antarctic firn, The Cryosphere, 5(4), 809–819, doi:10.5194/tc-5-809-2011, 2011.

Pritchard, H. D., Ligtenberg, S. R. M., Fricker, H. A., Vaughan, D. G., van den Broeke, M. R. and Padman, L.: Antarctic ice-sheet loss driven by basal melting of ice shelves, Nature, 484(7395), 502–505, doi:10.1038/nature10968, 2012.

Shean, D.: Quantifying ice-shelf basal melt and ice-stream dynamics using high-resolution DEM and GPS time series, Ph.D. Thesis, University of Washington, Seattle, WA, 14 July. [online] Available from: https://digital.lib.washington.edu:443/researchworks/handle/1773/36365 (Accessed 22 November 2016), 2016.

---

## Author Response (AR2)

Review #1

SUMMARY

This is my second review of this paper, which describes a GPS-based technique to make estimates of basal mass balance (BMB) and surface mass balance estimates for Pine Island Glacier ice shelf (PIGIS). The inclusion of GPS-derived snow surface height relative to the GPS antenna is a particularly interesting addition to standard ice-shelf GPS analyses, and the results are tied in well to other utilized datasets including the stereo-imagery of the DEM and the surface snow/firn state from a RACMO-based firn density model (FDM).

The paper is much clearer than the previous version, due largely to simplifying the set of vertical coordinate variables.

Specific comments are divided into "Major", where I'd like to know what you did in response, and "Minor", which don't require documentation in revision. Numbers refer to original page.line.

-- Laurie Padman

GENERAL COMMENTS

1. I think "PIG shelf" should always be "PIG ice shelf", in which case you might want to introduce "PIGIS" early and always use it.

We do not want to introduce additional acronyms and feel that "PIG shelf" is sufficient.

2. I don't like the constructions "between/from 2012-2014" as it requires reading the dash as "and" or "to". Better to replace the dash with a word, or say "the period 2012-2014". Similarly for ranges of dimensional values.

We changed most of these instances to "the period 2012–2014" but chose to leave some for simplicity. We consistently use the em-dash for ranges, which implies "to" in this context.

3. I'd try to avoid the construction "Figure X shows that …". It suggests that a pre-existing set of figures is driving the paper, rather than the figures supporting the facts. It's usually possible to say something like "The elevation at each site (Figure X) …"

This is stylistic and does not change the meaning of the text. We left as is.

MAJOR COMMENTS

3.20-3.22: You almost, but didn't quite, finish the mass balance statement for the ice shelf itself. If 95-101 GT/yr for basal melting is 70-80% of mass loss from the ice shelf, then calving is 20-30 Gt/yr (although Rignot et al. 2013 says 62 Gt/yr), and SMB is 5 (Rignot). Does that add up to give the observed thinning rate?

Part of the confusion on this point is related to numbers from different time periods and different techniques. The estimates from Rignot et al. 2013 have been refined by more recent studies as cited in the text. A more thorough discussion of these numbers is presented in [*Shean*, 2016], and will be published in forthcoming papers.

4.32-4.34: Both bedrock GPS sites are outside the domain for Figure 1, right? Are they good choices, and are there different penalties for the different sites? Regardless, maybe tell us the typical distance away from the ice shelf GPS for each site.

We included distances in an earlier version, but removed to simplify this portion of the text. We reinserted to accommodate the reviewer's request. The HOWN site is far from the PIG shelf, but it was the best available base station data for 2008-2010.

5.2-5.3: I don't understand why you subset the 30-s positions to 10-minute intervals: you're throwing away 95% of the data?

This was done to reduce data volume and does not impact our results. Given the horizontal velocities of ~10 m/day, the displacement of the receivers over 15-second intervals is negligible. Even 10-minute intervals could be considered overkill for some analyses over a 2-year period.

5.11-5.18: This question was asked by both reviewers last time around: Why use a tide model when your data contains the exact tide, and the records are long enough to analyze? I'd expand this to suggest that a better IBE correction might be possible following Padman et al. (2003) rather than just using 1 cm/hPa. The correlation of the height data with the pressure from ERA-Int or the AWS would give you a better model. If you continue to use CATS2008, then a better expression for citing it is "an updated version of the model described by Padman et al. [2002]"

We updated the CATS2008 citation text, and we do cite the Padman et al (2003) paper. As specified in our original response to reviewers, the suggestion to redo the entire analysis with a different tide model or tide removal approach would not change our results. We appreciate the reviewer's comments and will consider this for future work.

7.22 and elsewhere: I am not familiar with NMAD. (1) You need to tell us something about it (what does it tell you that RMS doesn't), and (2) does it have units? If so, why "normalized" ?

NMAD is a robust metric of variability, less affected by outliers than std or rms.
https://en.wikipedia.org/wiki/Median_absolute_deviation
For the normal distribution, the "normalized" MAD (i.e. MAD multiplied by constant scale factor of 1.4826) is consistent with the standard deviation.
This is a standard statistical metric and we do not feel it requires further discussion in the text.

8.4-8.8: (1) $v\_fc$ is fairly important in your study, but there's not much information here to understand it other than calling it "dry firn compaction". My possibly wrong interpretation is that it is the height change associated with firn compaction *below* the pole base, and that this rate is set by the amount of new SMB above the pole base. But this needs to be clearer, and I'd also

want to know how the choice of a fixed "firn air content" of 12 m relates to what v_fc can be. (2) related: How does all this get us to an estimate of uncertainty on v_fc?

We include references to [*Ligtenberg et al.*, 2011] which contains a much more thorough description of v_fc and its uncertainty. This is beyond the scope of the current paper.

v_fc can be estimated for any layer in the firn column. In this paper, we estimated v_fc for a layer at the depth of the pole base over time. The reviewer is incorrect – loading from new SMB above the pole base has limited influence on v_fc. Rather, v_fc is primarily controlled by compaction rates in the underlying firn column, which are related to cumulative SMB history.

Regardless, based on feedback from both reviewers, and further discussion with coauthors, we decided to remove the discussion of v_fc from the text.

9.3-9.8: This is confusing as, at first thought, the relationship of the pole base to the moving firn-ice transition is not obvious. I think I understand v_fc now (see previous comment), but then the assumption of constant firn air (12 m) creates a relationship between the firn-ice transition and the total mass content of the firn layer, implying that compaction for the firn layer depth range under the pole base is determined by recent precip above the pole base.

We again refer the reviewer to [*Ligtenberg et al.*, 2011], which contains a more detailed discussion of IMAU-FDM. The firn model simulations do not involve a moving firn-ice transition, and do not require constant firn air. The latter was introduced as a simplification for the basal melt rate derivations in Section 3, which we justify in the text "limited temporal variability (+/-0.3 m or ~1-3%) in modeled IMAU-FDM total firn air content for the three PIG shelf grid cells during the relevant ~2-year study periods."

In the model, additional surface accumulation/ablation will result in total firn column thickness change, which will involves surface elevation change above the firn-ice transition.

9.13-9.14: The density ratio term in eq. (7) explains why you say that BMB is 9 times more sensitive to z_surf than to adot. However, this ignores the scaling of div.u and the relationship between z_surf and adot. Set divergence to zero, on a flat ice shelf with no BMB, and z_surf relates to adot, but the divergence term doesn't even show up.

We are not ignoring div.u scaling, we are merely stating that "basal melt rates are ~9 times more sensitive to surface elevation change ($Dz_{surf}/Dt$) than SMB ($\dot{a}$) for a floating ice shelf". The divergence term is very small (as supported by observations presented in the text), but scaling of that term is also 9x larger than adot. As for the reviewer's comment about a flat ice shelf - if bdot is 0 and div.u is 0, then we are left with (Dz_surf/Dt)(~9) = adot.

10.28-10.29: Not "in the upper few meters of the firn column", just "in the firn column above the pole base". Right? What happens *below* the pole base doesn't affect the antenna to surface distance.

This is correct. The "upper few meters" and "above the pole base" are synonymous in this case. We reworded to clarify:

"All GPS array records show an abrupt antenna-surface distance increase (~0.2–0.3 m) between December 2012 and January 2013, which is consistent with surface melting and/or enhanced firn compaction rates above the pole base (i.e., upper few meters of the firn column). "

12.10-12.12: This seems a bit disingenuous. The signal is being interpreted entirely as though all the assumptions are correct. You explain later (Section 5) that there are reasons it might be wrong, especially Section 5.1, but the uncertainties need to be addressed briefly here.

The magnitude of the melt rate differences between these sets of receivers is significantly greater than the measurement uncertainty. We feel that the organization presenting basic observations in this section, and discussing the details of uncertainty in later sections is appropriate.

13.10-13.17: This is not very convincing (to me). My reading of your paper, without being steeped in the PIGIS literature, is that longitudinal extension might be biased towards transverse basal channels, and that hydrostatic equilibrium of short scales might be being slowly approached over the decade since the ice began to float. Can you really resolve the dynamic term, given the scales of the transverse rifts seen in the right-hand panels of Figure 10?

This section was added to the first revision to address the other reviewer's concerns about hydrostatic equilibrium for channels/keels with length scales of ~1-2 km. It is not intended to address the length scales of the transverse rifts brought up by the reviewer. These issues are addressed in Section 5.4 on "Strain rate length scales"

15.10-15.11: Maybe I don't understand this all well enough; but in what way is PIG2 SMB "in balance with" ongoing firn compaction and basal melt during this period? In mass, or height? It seems like the balance is that ~2 m per year of fresh firn is deposited, but Table 1 suggests BMB is ~2 or ~4.4 m/yr (depending on method) of ice, and BMB is only ~1 m.w.e. per year, so it's only a balance in terms of thickness. But then compaction is fixed to only allow 12 m of firn air at all times?

The full sentence was "The limited variability in surface elevation at PIG2 (Figure 7C) suggests that the observed 2008–2010 SMB over the South PIG shelf was approximately in balance with ongoing firn compaction and basal melt during this period."

So, "surface elevation" involves height. Essentially, we are saying Dz_surf/Dt is close to zero for these years, so other terms in Equation 5 must be equal. As stated earlier, compaction is not fixed and we removed the language "ongoing firn compaction," which should hopefully address the reviewers concern.

The modified text reads "The limited variability in surface elevation at PIG2 (Figure 7C) suggests that the observed 2008–2010 SMB over the South PIG shelf was approximately equal to basal melt during this period, assuming negligible velocity divergence for this location. "

15.16: You can't say "appear to be uncorrelated". You either mean just "unrelated" (or "causally unrelated"), or you calculate the correlation and decide if it passes a statistical threshold or not.

We said "appear uncorrelated" but did not make the direct claim that they were "statistically uncorrelated". Changed to "unrelated"

16.11-16.31: I recommend rolling these two subsections together, and starting with the discussion of spatial variability so that, when you compare with the two direct measures of BMB, you already have the justification explained.

We are satisfied with the current organization and feel these two sections should be separate.

16.33-17.3: "that appear to display a lagged …". If this is true, then show it. However, once you start the sentence "Our analysis …", you seem to be stepping away from accepting the lagged correlation with ocean T. Overall, you seem to set up a belief that the ocean matters and that you have evidence for it, but then say "Actually, no, it's something else."

We modified to clarify that the first sentence is the assertion of [*Christianson et al.*, 2016]:

"Christianson et al. [2016] suggest that the subtle (~2–4%) changes in 2012–2014 GPS velocity display a lagged correlation with observed variations in ocean temperature records from moorings in Pine Island Bay (see Figure 1 for location), potentially implying causality."

MINOR COMMENTS

1.27: "limited" is unnecessarily vague here.

We feel this is appropriate for the abstract, and present details in the text.

2.6: "relatively coarse grid". By most standards, these grids are fairly "fine". You need to tell us what grid you need, and why.

We do specify desired SMB resolution (<1-km) and why in Section 6.7. We do not feel this belongs in the introduction.

2.7-2.8: This sentence seems to imply that installing GPS (the topic of this paper) is easier than these are things, but the logistics are similar.

We are not implying that GPS is easier, merely stating that field installations are logistically challenging. The ability to extract new information from existing GPS installations expands available options for cal/val without additional fieldwork.

2.10: Why "cumulative" balance?

A small bias in seasonal or annual mass balance observations can lead to large cumulative errors over time.

2.22-2.23: dynamic firn models are forced by a lot more than just SMB.

The text does not imply that modeled SMB is the only forcing.

2.35: "temporally dense" seems complicated: Why not just say "continuous" ? Overall, the issue is whether the single-to-noise combined with sampling characteristics gives you more valuable results than other methods at time *and space* scales you want to resolve.

We changed "temporally dense" to "continuous" and keep the later sentence that states "This approach yields temporally dense records of basal melt rates at spatially sparse GPS locations…", which is the point we are trying to make.

3.16: This net mass loss (40-50 GT/yr) applies to the entire PIG grounded-ice catchment, right?

Yes, we modified to read "net mass loss estimates of 40 to 50 Gt/yr for the full PIG catchment".

4.2: Don't see the need to hyphenate "ice sheet" and "ice shelf" here

We hyphenated because both are used prior to "dynamics" (i.e., "ice-sheet dynamics" and "ice-shelf dynamics"). Defer to TC editorial staff.

4.27: As expanded upon, "fortuitous" seems like the opposite of what you mean!

Deleted "fortuitous"

4.29: I don't understand the '(co)'

Deleted "(co)"

5.8-5.9: What are the "8 km GPS paths"? Do you mean "receiver separations"?

No. We modified to clarify:

"Absolute geoid errors are poorly constrained for coastal Antarctica, but relative geoid error for the cumulative horizontal displacement of the GPS array (~8 km over the 2-year period) should be <1-2 cm"

6.30 (but check everywhere): Consistent use of italics for Lagrangian derivative D/Dt. Note that some oceanography texts would say D{\itX}/D{\itt}.

They are consistent. We defer to TC editorial staff for preferred formatting.

8.21: If rho_ice =917 +/- 5, then how can you use 917 as the threshold for identifying the height of the firn-ice transition? Doesn't that become a noisy estimate?

The +/-5 kg/m3 was not used with FDM, but is used for uncertainty calculations elsewhere in this paper (i.e., related to hydrostatic scaling term). We refer the reviewer to [*Ligtenberg et al.*, 2011] for details on the FDM use of 917 for this transition.

8.23: units for 'd \approx 12 *m*'

Good catch.  Changed.

9.24: The sign of the shear "dextral (right-handed)" doesn't tell me anything. All I care about is that velocity is higher towards the center of the trunk flow, right?

Yes, we present magnitude and specify direction as dextral.

10.9-10.10: I really don't like the format "increased (decreased) … increase (decrease)". First, it's hard to read. Second, it's obvious, right?

We are confident that most readers are familiar with this presentation, and defer to TC editorial staff.

No, this is not necessarily obvious.  If the entire array speeds up uniformly, there may not be any change in intra-network strain rates.

11.11: Italics for z_surf etc.

Again, good catch.  Changed.

11.19-11.20: I don't think you mean "scaled" temperatures; you mean "calibrated" temperatures, or local temperatures estimated from the relationship between PIGIS AWS and Evans Knoll.

Changed to "calibrated".

11.27-11.29: Would have preferred to see a graphic of the historical context for warm periods in a longer time range. Seems important, especially if it figures into total firn state.

This is beyond the scope of this manuscript, but we include the 1979–2016 ERA-Interim 2-m T record here for completeness.  Top panel shows full temperature scale, bottom panel shows zoomed y-limits of approx. +/- 2°C.

[Figure]

[Figure]

14.2: "We now consider the possibility that" => "It is possible that" ?

Changed to "We now consider whether…"

14.13: "Finally, we assume that …" Out of style with this section, which starts each para with a discussion fo what could go wrong, not what you assumed. So here, maybe "It is possible that the poles tilt over time."

We modified the introductory sentence to include all three potential issues:

"We now consider whether some of the observed $Dz_{surf}/Dt$ could be related to settling, heating, or tilting of the GPS poles over time."

14.33: My reading is that v_fc represents "firn compaction below the pole base", not total firn compaction.

v_fc is the downward velocity due to firn compaction for a layer in the firn column, in this case, the layer containing the pole base.

16.4-16.5: *Why* is there missing antenna-surface distance data during this period? What causes loss of this valuable information?

This was addressed in Section 2.3 "The SOW3 record was curtailed on August 22, 2013, when antenna-surface distance decreased below the minimum threshold of ~0.5 m (Nievinski, 2013)."

We added "(see Section **Error! Reference source not found.**)" here for clarification.

17.27: spell out "cal/val"

Changed to "calibration/validation"

17.30: "Set GPS elevation mask to 0^o". Only makes sense if you tell/remind us what the present setting is, and *why* that's the present setting; i.e., is there a penalty on other measurements when you change the mask specifically for antenna-to-surface heights?

Changed to "set the GPS elevation mask to 0° (default values are typically ~5–10°)"

Setting to 0° potentially introduces more multipath noise from low elevation satellites for position estimates, but this filter can also be applied during post-processing.

18.3: Does NMAD have units?

The text reads "NMAD of ~0.57 m". No change is necessary.

FIGURES

See RC1 for some other figure comments.

F.4: This figure would be a good place to define v_fc.

The first round of reviews suggested that this figure was too complicated, with too many variables.

We simplified the text and removed much of the discussion around v_fc, so no change is necessary.

F.6: What does "annualized velocity magnitude" mean, when it clearly isn't "annualized"?

This was an attempt to state that we scaled the 42-day displacement to m/yr. Removed "annualized" to avoid confusion.

F.7 and F.9: full-page-width would be good. Especially for Fig. 9, where the text asks us to see short-time-scale events in the 2012-2014 period.

Agreed. Will bring up with TC typesetting staff.

Review #2

The authors have gone to some effort to redraft the manuscript along the lines suggested by the reviewers. The result undoubtedly represents an improvement over the earlier version. The overall structure now seems easier to follow and the reduction in the number of variables has helped considerably. I also found the sections discussing the issue of isostatic equilibrium and stability of the poles useful. They remove many of the doubts from my mind about the results. However, in places I still found it hard to follow what the authors are actually trying to present.

1) The section I struggled with most was number 3, which apparently derives two expressions for the basal mass balance. The reason for using two expressions in never entirely clear, while the derivations are a little unsatisfactory.

In Section 3, we stated "The effects of processes that drive short-term surface-elevation change (e.g., accumulation, melting) are largely absent below the upper few meters of the firn column." In other words, z_surf includes variability due to SMB.  This is not the case for z_ant, which is useful for isolating subtle variability in the basal melt signal.  This is clear in Figure 9 A and B, and the caption included the sentence "Note limited residual magnitude and dampened seasonal signal of $z_{ant}$ compared to $z_{surf}$. Unlike $z_{surf}$, no significant change is observed in $z_{ant}$ from Dec. 2012 to Jan. 2013."

We felt this was clear, but we modified section 3 to address the reviewer's concern, and now present only one derivation for basal melt rates from surface elevation change.  This change also addresses many of the reviewers concerns below.

For equations (1) to (5), the authors make a conventional assumption that the thickness of air within the firn is constant and can simply be subtracted from the total thickness to leave a solid-ice-equivalent thickness. That is fine, but raises a couple of questions:

(i) In what sense is (4) an approximation? Is it the assumptions of constant density that are subsequently made, but if so, why is (5) exact?

Assumptions about densities, firn air content, and hydrostatic equilibrium all contribute to uncertainty in the ice thickness estimates.  We changed the $\approx$ to =, as the wording in revised manuscript is more explicit about assumptions for equation 5.

(ii) If it is OK to assume constant air content, which it appears to be, why worry about firn compaction at all? You could make the paper a whole lot simpler if you just left it at that. Is it considered because of the insight that the GPS records give into the process? If so, then it should really be given the same status as surface and basal mass balance in the title and abstract. It is an independent process that you are studying. The new section 3 now clarifies that you don't actually need it at all to derive basal mass balance.

As stated above, the GPS antenna elevation time series can potentially provide basal melt rate estimates without SMB variability in z_surf time series. To use z_ant, we need to account for

downward motion due to firn compaction (and other processes) in order to isolate the downward motion due to basal mass balance.

While we do feel that new information on firn compaction can be gleaned from the GPS records, we accepted the reviewers suggestion, and cut this from the paper. We hope to address this topic further in future work.

I struggled with equation (6). From Figure 4 it is clear that:

Z_ant = Z_surf + h_(ant-surf)

so the exact equation, from which the approximation in (6) is derived, has the total derivative of h_(ant-surf) on the right-hand side. That term is a combination of surface accumulation and compaction of the firn above the level of the (fixed) pole base.

This is a good point. The initial version of eq 6 was incorrect. We removed eq 6 from the paper.

This raises further questions:

(iii) What are the assumptions that you make to get to your version of (6)?

We removed eq 6 from the paper.

(iv) If these involve the assumption of steady accumulation and firn compaction, how is that distinct from the assumption of constant air content?

We removed eq 6 from the paper.

(v) Are equations (5) and (7) supposed to be independent? How can they be considered as such, when (7) still includes the surface elevation minus the air content?

z_surf is derived from two independent signals (z_ant and h_ant-surf). We compute linear fits to both observed Dz_ant/Dt and observed Dz_surf/Dt.

The previous eq 7 has been removed.

Overall, it would seem simpler to leave (5) as the source of the basal melt rates and discuss (6) in its exact and approximate forms in terms of the processes of firn compaction and how the model compares with the observation of the compaction velocity.

We maintain that there is value in considering basal melt rates derived from both Dz_surf/Dt and Dz_ant/Dt observations, but we implemented the reviewer's suggestion to limit basal melt rate derivation to eq 5, and leave most of the firn compaction and model evaluation discussion for a future paper.

2) The discussion of the derived melt rates is somewhat misleading. The authors mention the

50% reduction in melting reported by Dutrieux et al (Science, 2014), but those observations pre-date the GPS data discussed in this manuscript. I overlooked that in my earlier review, because I did not check back to the other papers. I should have done, but the text of this manuscript gave me no reason to suspect that the observations were not contemporaneous. I think that critical point should be clarified, and the relevance of those earlier observations should also be spelt out.

This is a good point - an oversight in an ongoing effort to split a long PhD thesis with a broader set of conclusions into discrete journal articles. The comparison with [*Dutrieux et al.*, 2014] belongs in a companion paper presenting a time series of annual DEMs used to derive 2008-2015 melt rates for the entire PIG ice shelf.

The exact date range for the 2012 observations in [*Dutrieux et al.*, 2014] is not specified in the manuscript or supplement (or we did not see dates upon review – presumably several different periods in Jan 2012?) The GPS records begin Jan 11, 2012, so presumably there is some overlap with the 2012 cruise observations. But not necessarily enough overlap to warrant a direct comparison.

We modified the text in Section 6.6 to address this oversight.

Are the authors assuming, based on Figure 2a of Christianson et al (GRL, 2016), that the variability through the observational period was as large as that before it? Maybe their results are telling them that the upper water column variability matters less for the melt rates?

We are only considering the variability during the periods when we have GPS observations.

In Webber et al (Nature Communications, 2017), the full ocean records are presented. In Figure 4 of that paper the eye is drawn to the upper water column variability that presumably dominates the average numbers in the Christianson et al figure. But deeper down you see a steady decline to January 2013, followed by a slight recovery. If temperatures at around 700 m are the critical factor, then most of the decline occurred in the period up to January 2012 (sampled by Dutrieux et al) and the variability through the period of the GPS records (discussed here) was much more muted. Could this explain the relatively steady melt rates observed? And how steady are they? Some of the lines seem to show a gradient that is reducing in the first part of the record, then increasing in the second part. Could that be consistent with slight cooling to 2013, followed by slight warming?

This is all valuable information and insight, but is beyond the scope of this manuscript. The GPS records tell us that basal melt rates at these sites did not vary during this period. Most of the interesting "why" questions would only involve further speculation on our part. We will take these useful comments into consideration as we finalize subsequent manuscripts on the subject.

The record that seems at odds with this interpretation is the altimeter record presented in Figure 2a of Christianson et al. Is that the same record as the one discussed here? Why is the temporal variability shown by Christianson et al now ignored?

Section 6.4 includes a discussion of the bottom altimeter data, and its limitations, which preclude a direct comparison (at least in the opinion of the first author).

3) Some more minor points:

(i) Page 1, Line 16: "To better understand …" Does the Cryosphere accept the split infinitive?

We defer to TC editorial team.

(ii) Page 3, Line 19: I don't think I would describe the ice shelf as "large"!

Changed to "…terminates in an ice shelf ("main shelf")…"

(iii) Page 5, Line 28: Here you mention firn compaction above the pole base, but I don't think you ever quantify this term do you? When you use the term v_cf, is it always compaction below the pole base, or is it sometimes the compaction over the whole firn column? It would help if you were clearer on this point.

v_fc can be estimated for any layer.  Throughout the paper, we used v_fc as the downward velocity of a tracer at the pole base.  Regardless, we removed most discussion of v_fc to reduce confusion.

(iv) Page 7, Line 5: Shouldn't it be the "Regional Antarctic Climate Model".

No, the text is correct.

(v) Section 6.1, paragraph 1: Here you need to be really careful about what you mean by firn compaction. You discuss the movement of Z_surf relative to its initial level. That is a function of surface accumulation and compaction above the initial level, isn't it? But you seem to use a compaction velocity determined for the firn below the pole base? I'm afraid I don't follow this.

As outlined in the response to reviewer comments submitted with revision v1 (See Figure 2), we also considered downward velocity of the z_surf0' tracer over time.  This is different than the v_fc for the pole base tracer.  This accounts for the effects of surface accumulation and near-surface compaction rates between z_surf0' and pole base layers.

(vi) Page 17, Line 19: Another split infinitive ("to further constrain").

We defer to TC editorial team.

(vii) Page 18, Lines 4-9: Again, why should Z_surf-Z_surf0 be affected by firn compaction below the pole base? Both elevations change at the same rate as a result of deep compaction. Surely, only compaction between the two levels can affect the elevation difference?

This is a good point.  We removed the reference to v_fc, so the text now reads:

"Surface elevation relative to a firn layer tracer for the initial surface ($z_{surf}$ - $z_{surf0}$') increased at rates of ~0.8–1.1 m/yr for all GPS sites, which is consistent with modeled SMB of ~0.7–0.9 m w.e./yr and modeled downward firn-compaction velocities"

(xi) Page 18, Line 21: See comments under 2), above. How can you be so sure that the temperature variability was significant?

Changed to "substantial" although the observed changes are "significant" given the measurement uncertainty of the mooring T sensors.

[revised manuscript text omitted]

$\dot{b}$ (m/yr)
from

**Page 24: [3] Deleted**         **David Shean**         **9/22/17 12:31:00 PM**

from $Dz_{ant}/Dt$